# Map7D2 and Map7D1 facilitate microtubule stabilization through distinct mechanisms in neuronal cells

Koji Kikuchi[1] , Yasuhisa Sakamoto[1], Akiyoshi Uezu[2], Hideyuki Yamamoto[3], Kei-ichiro Ishiguro[4] , Kenji Shimamura[5], Taro Saito[6], Shin-ichi Hisanaga[6] , Hiroyuki Nakanishi[1]

Microtubule (MT) dynamics are modulated through the coordinated action of various MT-associated proteins (MAPs). However, the regulatory mechanisms underlying MT dynamics remain unclear. We show that the MAP7 family protein Map7D2 stabilizes MTs to control cell motility and neurite outgrowth. Map7D2 directly bound to MTs through its N-terminal half and stabilized MTs in vitro. Map7D2 localized prominently to the centrosome and partially on MTs in mouse N1-E115 neuronal cells, which expresses two of the four MAP7 family members, Map7D2 and Map7D1. Map7D2 loss decreased the resistance to the MT-destabilizing agent nocodazole without affecting acetylated/detyrosinated stable MTs, suggesting that Map7D2 stabilizes MTs via direct binding. In addition, Map7D2 loss increased the rate of random cell migration and neurite outgrowth, presumably by disturbing the balance between MT stabilization and destabilization. Map7D1 exhibited similar subcellular localization and gene knockdown phenotypes to Map7D2. However, in contrast to Map7D2, Map7D1 was required for the maintenance of acetylated stable MTs. Taken together, our data suggest that Map7D2 and Map7D1 facilitate MT stabilization through distinct mechanisms in cell motility and neurite outgrowth.

## Introduction

Microtubule (MT) dynamics play crucial roles in a variety of cellular processes, including mitosis, and vesicle/organelle transport, as well as cell motility and morphology (Etienne-Manneville, 2013; Roll-Mecak, 2020; Cleary & Hancock, 2021). MT dynamics are altered in response to various intrinsic or extrinsic signals and are then modulated through the coordinated actions of various MT-associated proteins (MAPs), which control the processes of dynamic instability (Roll-Mecak, 2020; Cleary & Hancock, 2021). Therefore, it is important to identify and characterize MAPs to understand the regulatory mechanisms of MT dynamics. We previously performed a comprehensive proteomic analysis of MT co-sedimented proteins from the brain and identified a series of functionally uncharacterized MT-binding proteins (Sakamoto et al, 2008). The list included MAP7 family members Map7, Map7D1, and Map7D2 but not Map7D3. Among the MAP7 family, Map7 has been extensively characterized. Several lines of evidence suggest that Map7 has the ability to stabilize and reorganize MTs. Ectopic expression of Map7 induces MT bundling and resistance to nocodazole treatment-induced MT depolymerization (Masson & Kreis, 1993). Map7 expression is up-regulated during MT reorganization in response to the differentiation of keratinocytes (Fabre-Jonca et al, 1999) and the establishment of apicobasal polarity in human colon adenocarcinoma cell lines, including Caco-2 and HT-29-D4 cells (Masson & Kreis, 1993; Carles et al, 1999). In addition, Map7 and the *Drosophila* Map7 homolog ensconsin (Ens) are involved in kinesin-1–dependent transport by promoting the recruitment of a conventional kinesin-1, Kif5b, and its *Drosophila* homolog, Khc, to MTs during various biological processes (Sung et al, 2008; Metzger et al, 2012; Barlan et al, 2013; Kikuchi et al, 2018; Tymanskyj et al, 2018; Hooikaas et al, 2019). The competition between Map7 and other MAPs for MT binding regulates the loading of motor proteins, thereby controlling the distribution and balance of motor activity in neurons (Monroy et al, 2018, 2020). A considerable body of evidence has highlighted the important roles of Map7 in the regulation of MT dynamics. Similar to Map7, the MAP7 family member Map7D2 facilitates kinesin-1–mediated transport in the axons of hippocampal neurons (Pan et al, 2019); however, the function of Map7D2 in the regulation of MT dynamics and its relationship with other MAP7 family members remain unclear.

In this study, we first determined the tissue distribution and biochemical properties of Map7D2 in detail. Map7D2 is expressed predominantly in the glomerular layer of the olfactory bulb and the Sertoli cells of testes. Furthermore, it directly associates with MTs through its N-terminal half, similarly to Map7, significantly enhancing MT stabilization. We also examined the cellular functions of

[1]Department of Molecular Pharmacology, Graduate School of Medical Sciences, Kumamoto University, Kumamoto, Japan [2]Department of Cell Biology, Duke University Medical School, Durham, NC, USA [3]Department of Biochemistry, Graduate School of Medicine, University of the Ryukyus, Okinawa, Japan [4]Department of Chromosome Biology, Institute of Molecular Embryology and Genetics, Kumamoto University, Kumamoto, Japan [5]Department of Brain Morphogenesis, Institute of Molecular Embryology and Genetics, Kumamoto University, Kumamoto, Japan [6]Department of Biological Sciences, Tokyo Metropolitan University, Hachioji, Japan

Correspondence: kojik@kumamoto-u.ac.jp

Map7D2 using the N1-E115 mouse neuroblastoma cell line, which expresses both Map7D2 and Map7D1 but not Map7 nor Map7D3. Map7D2 predominantly localizes to the centrosome and partially on MTs and suppresses cell motility and neurite outgrowth by facilitating MT stabilization via direct binding. Finally, we determined the functional differences between Map7D2 and Map7D1 with regard to MT stabilization in N1-E115 cells. Although Map7D1 exhibits similar subcellular localization and gene knockdown phenotypes to Map7D2, Map7D1 is required to maintain the amount of acetylated tubulin in contrast to Map7D2. These results suggest that Map7D2 and Map7D1 facilitate MT stabilization through distinct mechanisms for the control of cell motility and neurite outgrowth.

## Results

### Map7D2 is highly expressed in the glomerular layer of the olfactory bulb and the Sertoli cells of testes

In our previous comprehensive proteomic analysis of co-sedimented MT proteins from the rat brain, we identified a number of novel proteins (Sakamoto et al, 2008). In the present study, we focused on Map7D2, which is one of the MAP7 family members (Fig S1A). Although the tissue distribution of Map7 has been well-analyzed by using mice (Fabre-Jonca et al, 1998; Komada et al, 2000), that of Map7D2 has not been analyzed. To analyze the tissue distribution of Map7D2, we first performed Northern blotting analysis using total RNA extracted from various rat tissues. Northern blotting analysis showed that the ~4.2-kb mRNA was hybridized only in the brain and testis, being more abundant in the former (Fig 1A). Of note, no detectable signal was observed in other rat tissues examined, including the heart, spleen, lung, liver, skeletal muscle, and kidney.

Next, we investigated the tissue distribution of Map7D2 at the protein level by immunoblotting. For the immunoblotting analysis, we raised an anti-Map7D2 polyclonal antibody using aa 1–235 of rat Map7D2 (rMap7D2) as an epitope (Fig S2A). Using lysates from HeLa cells transfected with an empty vector, hMap7-V5His$_6$, or rMap7D2-V5His$_6$, we confirmed that the antibody detected Map7D2 but not Map7 (Fig S2A). In addition, we evaluated antibody specificity by siRNA-mediated knockdown of endogenous Map7d2. For this experiment, we used a mouse neuroblastoma cell line, N1-E115, in which the expression of Map7d2 and Map7d1, but not Map7 and Map7d3, was detected by quantitative real-time PCR (RT-qPCR) (Fig S2B). We designed three independent siRNAs against Map7d2 or Map7d1. The immunoreactive band disappeared after treatment with each Map7d2 siRNA but not the control or Map7d1 siRNA (Fig S2C), indicating that the antibody specifically recognized Map7D2. We then performed immunoblotting analysis using lysates from various rat tissues. Consistent with the Northern blotting analysis, Map7D2 was detected at the protein level only in the brain and testis, whereas no immunoreactive bands were detected in other rat tissues (Fig 1B). In contrast to Map7D2, Map7 is widely expressed in a variety of mouse organs that contain epithelium (Fabre-Jonca et al, 1998; Komada et al, 2000). These data suggest that the mechanism of expression regulation may vary between Map7D2 and Map7.

We further analyzed the expression patterns of Map7D2 in the brain and testis by immunofluorescence. Based on RNA-seq CAGE, RNA-Seq, and SILAC database analysis (Expression Atlas, https://www.ebi.ac.uk/gxa/home/), Map7D2 expression was detected in the cerebellum, hippocampus, and olfactory bulb and not in the cerebral cortex (Fig S3). We confirmed Map7D2 expression in the above four brain tissue regions of postnatal day 0 mice by immunofluorescence. Among these regions, Map7D2 was the most highly expressed in the Map2-negative area of the olfactory bulb, i.e., the glomerular layer (Fig 1C). Weak signals were detected in the cerebellum, and marginal signals were observed in the hippocampus and cerebral cortex (Fig 1C). In contrast to Map7D2, Map7 has not been reported to be expressed in the glomerular layer of the olfactory bulb, whereas in the nervous system, Map7 is strongly expressed in the ganglia (Fabre-Jonca et al, 1998; Komada et al, 2000). Next, we analyzed Map7D2 expression in the seminiferous tubules of adult mice. Map7D2 signals were merged with Tubb3 signals, a marker for Sertoli cells (Fig 1C), indicating that similar to Map7 (Komada et al, 2000), Map7D2 is expressed predominantly in Sertoli cells. Taken together, these data suggest that in vivo, Map7D2 may function in the glomerular layer of the olfactory bulb and the Sertoli cells of the testis.

### Map7D2 has an ability to stabilize MTs

MAP7 family members share two conserved regions (Fig S1A–C). The amino acid sequences of the N-terminal (aa 53–138) and C-terminal (aa 389–562) regions of human Map7D2 (hMap7D2) were 64.0% and 42.9% identical to those of human Map7 (hMap7), respectively, whereas other regions showed no significant homology to hMap7 (Fig 2A). Using the rat brain cDNA library, we obtained rMap7D2 cDNA by PCR. The cloned cDNA encoded a protein consisting of 763 aa with a molecular weight of 84,823 (DDBJ/EMBL/GenBank accession number AB266744) (Fig 2A). The full-length aa sequence of rMap7D2 was 68.1% identical to that of hMap7D2. For subsequent experiments, we used the rMap7D2 that we cloned.

We sought to determine whether rMap7D2 directly binds to MTs. To this end, we performed an MT co-sedimentation assay using recombinant rMap7D2. When GST-rMap7D2 was incubated with MTs, followed by ultracentrifugation, it was recovered with MTs in the pellet (Fig 2B). The dissociation constant (Kd) was calculated to be ~6 × 10$^{-7}$ M (Fig 2B). This value is comparable to those of the well-known MAPs Tau and CLIP-170 (Gustke et al, 1994; Lansbergen et al, 2004). The stoichiometry of GST-rMap7D2 binding to tubulin was calculated to be one GST-rMap7D2 molecule per about 10 tubulin $\alpha/\beta$ heterodimers. This value was also comparable to that of Map7 (Bulinski & Bossler, 1994). It has been reported that Map7 binds to MTs through a conserved region on the N-terminal side, whereas Map7D3 binds via a conserved region on not only the N-terminal but also the C-terminal sides (Sun et al, 2011; Yadav et al, 2014). To further examine the location of the MT-binding domain of rMap7D2, the N-terminal (aa 1–421) and C-terminal (aa 422–763) halves were subjected to an MT co-sedimentation assay (Fig 2A). The N-terminal half was co-sedimented with MTs, whereas the C-terminal half was not (Fig 2C). These results indicate that the MT-binding domain of rMap7D2 is located only at the N-terminal half, similarly to that of Map7, but not Map7D3. Meanwhile, a region within the C-terminus

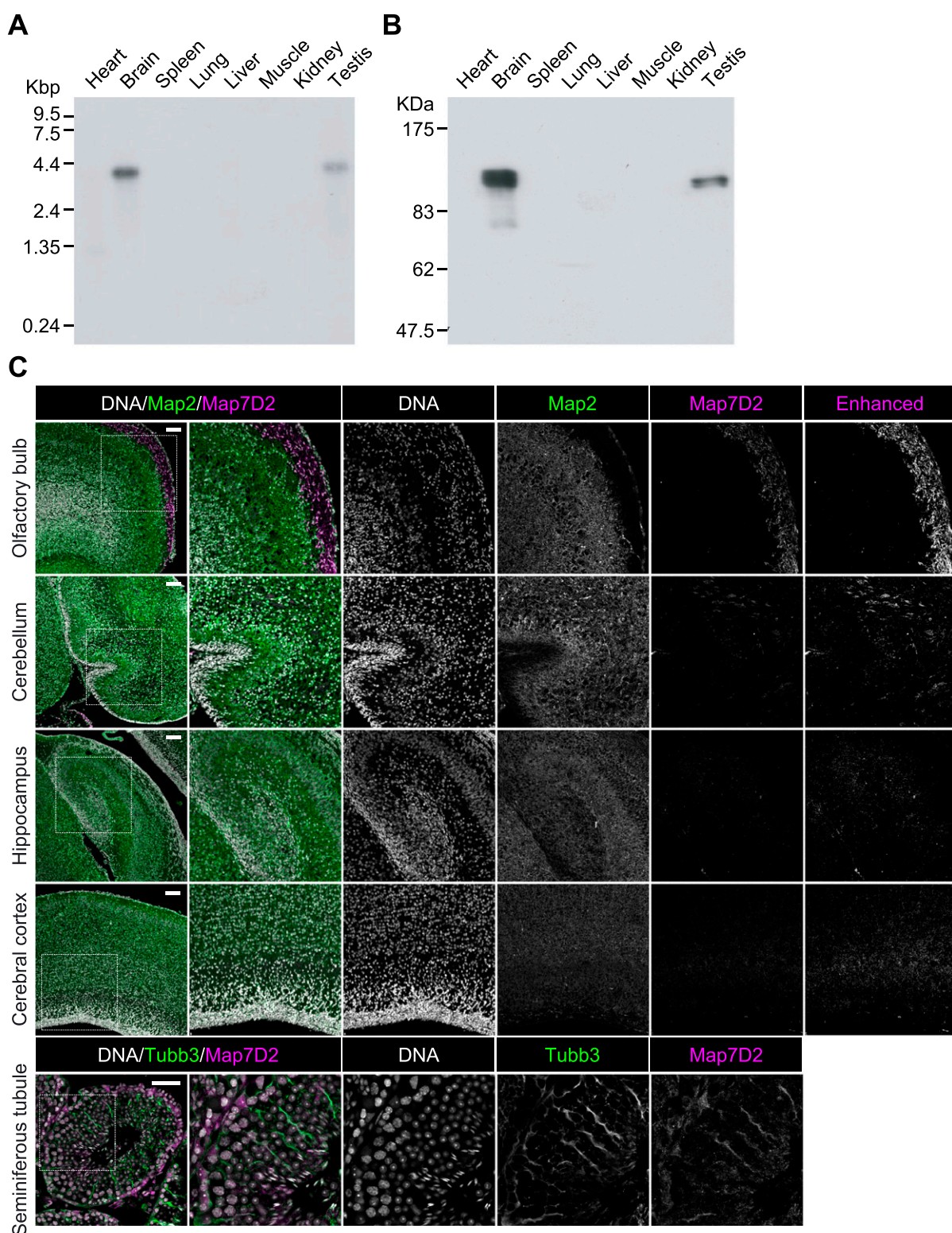

**Figure 1. Tissue distribution of Map7D2.**
**(A)** The full scan image of Northern blot analysis. An RNA blot membrane (CLONTECH) was hybridized with the [32]P-labeled full coding sequences of rMap7D2 according to the manufacturer's protocol. **(B)** The full scan image of Immunoblotting analysis. Various tissue lysates (20 μg of protein) were subjected to SDS–PAGE, followed by immunoblotting with the anti-Map7D2 antibody. **(C)** Expression patterns of Map7D2 in the brain and testis by immunofluorescence. Upper panels, frozen sagittal sections of postnatal day 0 mouse brains were stained with anti-Map7D2 (magenta) and antibodies against mature neuron marker Map2 (green). DNA was labeled with DAPI (gray). For a comparison of signal intensities, images were captured under the same parameters. Contrast-enhanced images of Map7D2 staining were shown in the rightmost

of Map7 is required for complex formation with Kif5b, the heavy chain of kinesin-1 (Fig S4), and is involved in Kif5b-dependent transport by loading Kif5b onto MTs (Metzger et al, 2012; Kikuchi et al, 2018; Tymanskyj et al, 2018; Hooikaas et al, 2019). Map7D2 also formed a complex with Kif5b (Fig S4), as previously reported (Pan et al, 2019). Taken together, these data suggest that the biochemical properties are largely conserved between Map7D2 and Map7.

Next, we tested whether rMap7D2 affects MT assembly. The MT turbidity assay was used to analyze the effect of rMap7D2 on the kinetics of MT assembly. The addition of rMap7D2 significantly enhanced the amount of polymerized MTs in a time-dependent manner, whereas tubulin self-polymerized even in the absence of rMap7D2 (Fig 2D). Identical results were observed by fluorescence microscopy analysis using rhodamine-labeled tubulin (Fig 2E). Furthermore, we investigated the ability of Map7D2 to bundle MTs in HeLa cells. Consistent with the in vitro data, overexpression of Myc-rMap7D2 induced MT bundling in HeLa cells (Fig 2F). Taken together, these results indicate that Map7D2 facilitates MT stabilization.

## Map7D2 localizes prominently to the centrosome and partially to MTs

After the biochemical characterization of Map7D2, we sought to determine its functions within the cell. To this end, we used N1-E115 cells that express Map7D2 and Map7D1 (Fig S2B and C). First, we analyzed the subcellular localization of Map7D2 in N1-E115 cells. N1-E115 cells can undergo neuronal differentiation in response to DMSO under conditions of serum starvation (Kimhi et al, 1976) and most of the cells extend neurites up to 12 h after treatment with 1% DMSO (Fig S5A) (Smit et al, 2003). In both proliferating and differentiated cells, Map7D2 localized prominently to the centrosome and partially to MTs (Fig 3A–C). These localizations were confirmed in N1-E115 cells stably expressing EGFP-rMap7D2 (Fig 3D and E). Furthermore, during cytokinesis, Map7D2 accumulated at the midbody, where MT bundling occurs (Fig 3B). Similarly, localization of Map7D2 was also observed at neurites, where MT bundling is also known to occur (Fig 3C). Together with the biochemical properties, these subcellular localization data suggest that Map7D2 is involved in MT stabilization within the cell.

As N1-E115 cells express another Map7 family member, Map7D1, we also determined its subcellular localization. Map7D1 exhibited similar localization to that of Map7D2 in both proliferative and differentiated states (Fig S5B–D). In the previous analysis using HeLa cells (Kikuchi et al, 2018; Hooikaas et al, 2019), Map7D1 localizes to MTs, especially from the vicinity of the centrosome toward the MT plus-end, with gradual weakening. This pattern is slightly different from the localization pattern in N1-E115 cells, suggesting that the mechanisms regulating localization patterns of Map7D2 and Map7D1 may differ depending on the cell type. Interestingly, *Map7d1* knockdown up-regulated Map7D2 expression, as confirmed with three different siRNAs (Fig S2C), suggesting that Map7D2 expression is affected by changes in Map7D1 expression, not by off-target effects of a particular siRNA. Moreover, the amount of

endogenous Map7D2 was decreased to 53% in N1-E115 cells stably expressing EGFP-rMap7D2 (Fig 3D), suggesting that EGFP-rMap7D2 expression suppresses endogenous Map7D2 expression. In this cell line, the total amount of Map7D2 was increased; however, when EGFP-rMap7D2 was depleted using si*gfp* in this cell line, endogenous Map7D2 was expressed to the same level as EGFP-rMap7D2 before knockdown (Fig 3D). Together with the finding that *Map7d1* knockdown increased the amount of Map7D2, these findings indicate that the amount of Map7D2 in the cells is regulated in response to the amount of Map7D1 and exogenous Map7D2. In contrast, *Map7d2* knockdown did not affect Map7D1 expression (Fig S2C), and identical results were observed in the Map7d2 knockout (*Map7d2*−/−) N1-E115 cells we generated (Fig S6A and B). As Ma7D2 expression was up-regulated upon suppression of Map7D1 expression, Map7D2 has the potential to functionally compensate for Map7D1 loss. Therefore, we decided to analyze the phenotypes of single and double knockdowns for *Map7d2* and *Map7d1* in the following experiments.

## Map7D2 and Map7D1 facilitate MT stabilization through distinct mechanisms for the control of cell motility and neurite outgrowth

Our biochemical analyses led to the possibility that Map7D2 is involved in MT stabilization within the cell. We initially analyzed the effects of *Map7d2* or *Map7d1* knockdown on the resistance to nocodazole, a MT-destabilizing agent. In control N1-E115 cells, a small degree of MT shrinkage was observed, and elongated MT arrays were retained even after treatment with a low concentration of nocodazole for 1 h (Fig 4A). In contrast, not only *Map7d2* but also *Map7d1* knockdown dramatically increased MT shrinkage when subjected to nocodazole treatment at the same concentration (Fig 4A), indicating that both Map7D2 and Map7D1 are required for MT stabilization within the cell. Furthermore, to investigate the possibility that Map7D2 or Map7D1 is involved in MT elongation, we measured the amount of EB1-decorated MTs at the cell periphery based on the intensity of EB1. The knockdown of either *Map7d2* or *Map7d1* did not affect the intensity of EB1 at the cell periphery, compared with control N1-E115 cells (Fig 4B), suggesting that both Map7D2 and Map7D1 are dispensable for the elongation of EB1-decorated MTs. Because Map7D2 and Map7D1 can form a complex with Kif5b (Fig S4) (Kikuchi et al, 2018; Pan et al, 2019), we also examined whether *Map7d2* or *Map7d1* knockdown affects the distribution of Kif5b foci. Even after *Map7d2* or *Map7d1* knockdown, the distribution of Kif5b foci was similar to that in control N1-E115 cells (Fig 4B). Kif5b foci were predominantly located at the internal regions of the cell (Fig 4B), and some were partly observed in the protrusions (Fig S7A). Taken together, these data indicate that Map7D2 and Map7D1 primarily stabilize MTs in N1-E115 cells.

The acetylation and detyrosination of α-tubulins are associated with stable MTs (Baas et al, 2016; Janke & Montagnac, 2017). Therefore, we examined the effects of *Map7d2* or *Map7d1* knockdown on the levels of acetylated and detyrosinated tubulins. Neither *Map7d2* knockdown nor knockout affected the total levels

---

column. Lower panels, frozen coronal sections of adult mouse testis were stained with anti-Map7D2 (magenta) and antibodies against Sertoli cell marker Tubb3 (green). DNA was labeled with DAPI (gray). Data information: In (C), scale bars in upper panels or lower panels are represented as 100 or 50 μm, respectively.

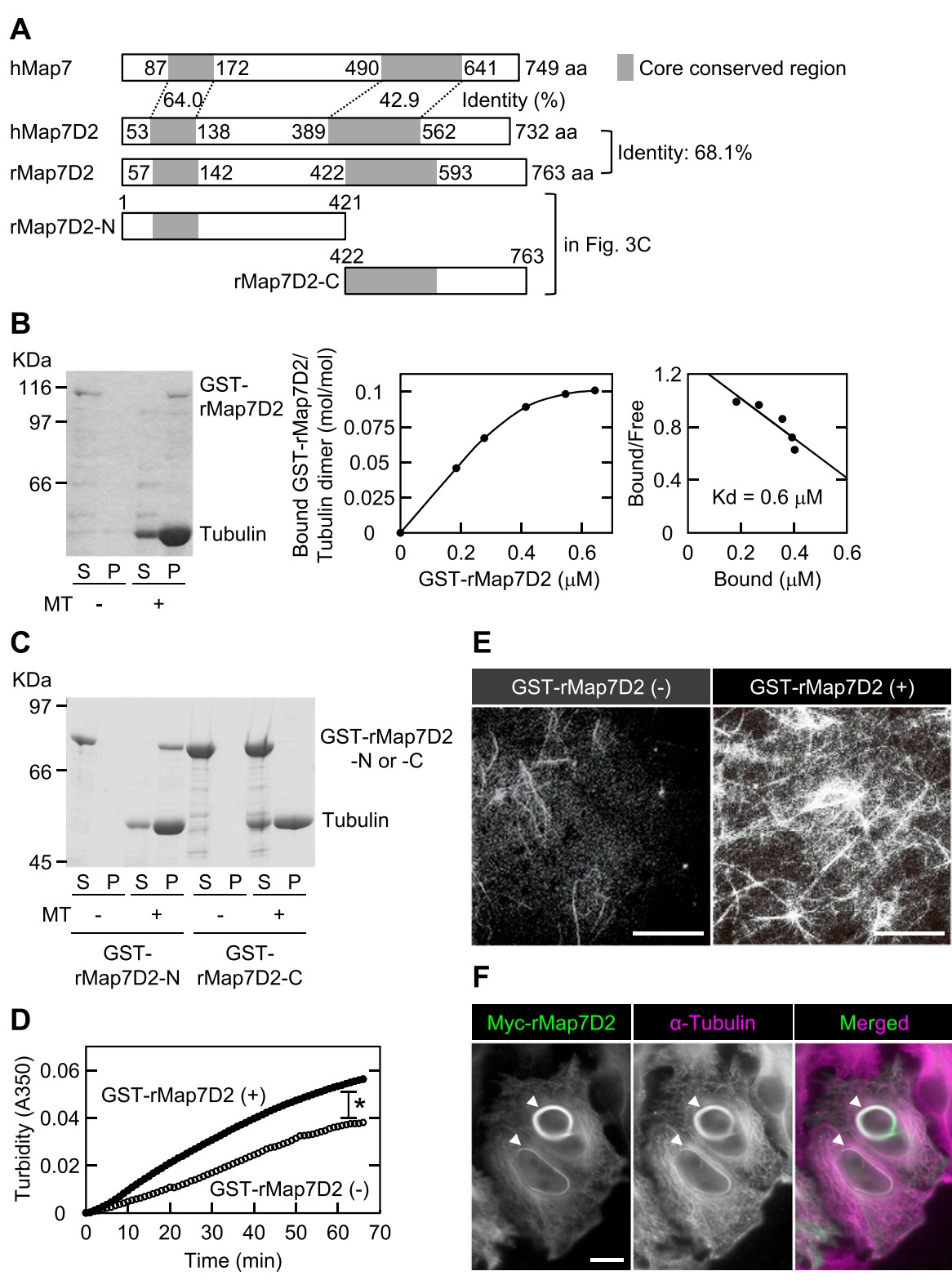

**Figure 2. Map7D2 has the ability to stabilize MTs.**
**(A)** Schematic structures of hMap7, hMap7D2, and rMap7D2. **(B)** Co-sedimentation of rMap7D2 with MTs. Left panel, GST-rMap7D2 (34 $\mu$g/ml) was mixed with MTs, followed by ultracentrifugation. Comparable amounts of the supernatant and pellet fractions were subjected to SDS–PAGE, followed by CBB protein staining. S, supernatant; P, pellet. Middle panel, various amounts of GST-rMap7D2 were mixed with MTs, followed by ultracentrifugation. Amounts of free and bound GST-rMap7D2 were calculated by determining protein amounts from the supernatant and pellet fractions, respectively, with a densitometer. Right panel, Scatchard analysis. **(C)** Location of the MT-binding domain. GST-rMap7D2-N (80 $\mu$g/ml) or GST-rMap7D2-C (200 $\mu$g/ml) was mixed with MTs, followed by ultracentrifugation. Comparable amounts of the supernatant

of these modified tubulins (Fig 5A and B). In contrast, *Map7d1* knockdown reduced the total level of acetylated but not detyrosinated tubulin (Fig 5A), and double knockdown of *Map7d2* and *Map7d1* had the same effect (Fig 5A). Consistent herewith, immunostaining revealed that *Map7d1* knockdown greatly decreased the intensity of acetylated tubulin around the centrosome in N1-E115 cells (Fig 5C and D). *Map7d1* knockdown decreased the intensity of α-tubulin and increased that of Map7D2 (Fig 5C and D), indicating that Map7D1 is required for the maintenance of acetylated, stable MTs. Under *Map7d2* knockdown, the intensity of α-tubulin and Map7D1 decreased without affecting that of acetylated tubulin (Fig 5C and D). This decrease in the intensity of Map7D1 is presumably because of a reduction in the number of MT structures that can serve as scaffolds for Map7D1 because the total amount of Map7D1 in the cells is not affected by *Map7d2* knockdown or knockout (Fig 5A and B). Together with our biochemical data for Map7D2, these results suggest that Map7D2 facilitates MT stabilization via direct binding, in contrast to Map7D1.

Dysregulation of MT stabilization affects various biological functions. For instance, it can lead to increased cell motility and neurite outgrowth (Biernat et al, 2002; Hubbert et al, 2002; Grenningloh et al, 2004; Alesi et al, 2016). Therefore, we analyzed whether random cell migration or neurite outgrowth of N1-E115 cells would be affected by single or double knockdown of *Map7d2* and *Map7d1* and *Map7d2* knockout. As expected, each single knockdown and *Map7d2* knockout enhanced not only the migration speed and distance during random cell migration (Fig 6A–C) but also the rate of neurite outgrowth (Figs 6D and S7B). Furthermore, double knockdown of *Map7d2* and *Map7d1* tended to result in increased cell motility and neurite outgrowth, compared with each single knockdown (Fig 6B–D). Taken together, these results suggest that Map7D2 and Map7D1 facilitate MT stabilization through distinct mechanisms, thus controlling cell motility and neurite outgrowth.

## Discussion

In the present study, we provide the comprehensive analysis of Map7D2 biochemical properties (Fig 2). The N-terminal and C-terminal regions of Map7D2 exhibited high homology to those of Map7 (Fig S1). The N-terminal homologous region is basic and highly charged. Most MT-binding domains characterized thus far are confined to positively charged regions (Lewis et al, 1989; Noble et al, 1989; Aizawa et al, 1990; Pierre et al, 1992). Consistently, the MT-binding region of Map7 was shown to be located at the N-terminal positively charged region (Masson & Kreis, 1993). Because we demonstrated that the N-terminal half of rMap7D2 directly bound to MTs (Fig 2C), it is likely that Map7D2 also associates with MTs through the positively charged N-terminal region. On the other

hand, the C-terminal region of Map7 includes the Kif5b binding domain (Metzger et al, 2012), which mediates Kif5b-dependent transport by loading Kif5b onto MTs (Metzger et al, 2012; Kikuchi et al, 2018; Tymanskyj et al, 2018; Hooikaas et al, 2019). Consistent with the conservation of this region between Map7 and Map7D2, Map7D2 also has the ability to form a complex with Kif5b (Fig S4) and contributes to kinesin-1–mediated transport in the axons of hippocampal neurons (Pan et al, 2019). Therefore, the biochemical properties of Map7D2 and Map7 are largely similar.

In contrast, the cellular functions of Map7D2 may differ from those of Map7. Our group and Hooikaas et al (2019) have previously reported that Map7 and Map7D1 have functional overlaps in HeLa cells (Kikuchi et al, 2018; Hooikaas et al, 2019). For instance, both form a complex with Dishevelled, a mediator of Wnt5a signaling, whereas Map7D2 does not (Kikuchi et al, 2018). In addition, Map7D2 exhibits distinct localization patterns in cultured hippocampal neurons, localizing to the proximal axon (Pan et al, 2019). In the present study, we propose a molecular mechanism explaining how Map7D2 and Map7D1 regulate MT stabilization in N1-E115 cells (Fig 6E). Map7D2 and Map7D1 both strongly localize to the centrosome and partially on MTs in proliferating and in differentiated N1-E115 cells (Figs 3 and S5B–D). Further, the knockdown of either resulted in a comparable reduction of MT stabilization (Figs 4A and 5C and D), without affecting the amount of EB1-decorated MTs (Fig 4B). Mechanistically, Map7D1 is required for the maintenance of MT acetylation, which is enriched in stable MTs, whereas Map7D2 is not (Fig 4). Although the above phenotypes exhibited by each knockdown are thought to enhance the rate of cell motility and neurite outgrowth, it is also possible that Map7D2 and Map7D1 control cell motility and neurite outgrowth through their function as complexes with Kif5b (Fig S4) (Kikuchi et al, 2018; Pan et al, 2019). However, it has been reported that when kinesin-1 function is impaired, both cell motility and neurite outgrowth are reduced (Lu et al, 2013; Agarwal et al, 2019). In addition, even after the knockdown of either *Map7d2* or *Map7d1*, the distribution of Kif5b foci was similar to that in control N1-E115 cells (Figs 4B and S7A). Although it is possible that in *Map7d2* or *Map7d1* knockdown N1-E115 cells, the effects of reduced MT stabilization are offset by those of kinesin-1 dysfunction, resulting in a mild increase in the rate of cell motility and neurite outgrowth, the phenotypes we observed are likely independent of the functions associated with Kif5b in N1-E115 cells. Alternatively, the fact that no stronger phenotype was observed may be because, besides Map7D2 and Map7D1, other molecules are involved in MT stabilization. Taking these findings and our biochemical data into consideration, we propose that, in contrast to Map7D1, Map7D2 facilitates stabilization by directly binding MTs, eventually to control cell motility and neurite outgrowth.

Slight differently from the localization of Map7D1 in HeLa cells (Kikuchi et al, 2018; Hooikaas et al, 2019), Map7D2 and Map7D1

and pellet fractions were subjected to SDS–PAGE, followed by CBB protein staining. S, supernatant; and P, pellet. **(D)** Turbidity measurement. GST-rMap7D2 was mixed with tubulin. The sample was incubated at 37°C and continuously monitored at 350 nm using a spectrophotometer. (○) without GST-rMap7D2; and (●) with GST-rMap7D2. **(E)** Immunofluorescent observation. GST-rMap7D2 was incubated for 20 min at 37°C with rhodamine-labeled tubulin. After fixation, the sample was spotted on a slide glass and viewed under a fluorescence microscope. **(F)** HeLa cells transiently overexpressing Myc-rMap7D2. Myc-rMap7D2 was transfected into HeLa cells, and the cells were then double-stained with anti-Myc and anti–α-tubulin antibodies. Arrowheads show MT bundles. Data information: In (D), *P < 0.003 (the F-test). Scale bars, 50 μm in (E) and 10 μm in (F).
Source data are available for this figure.

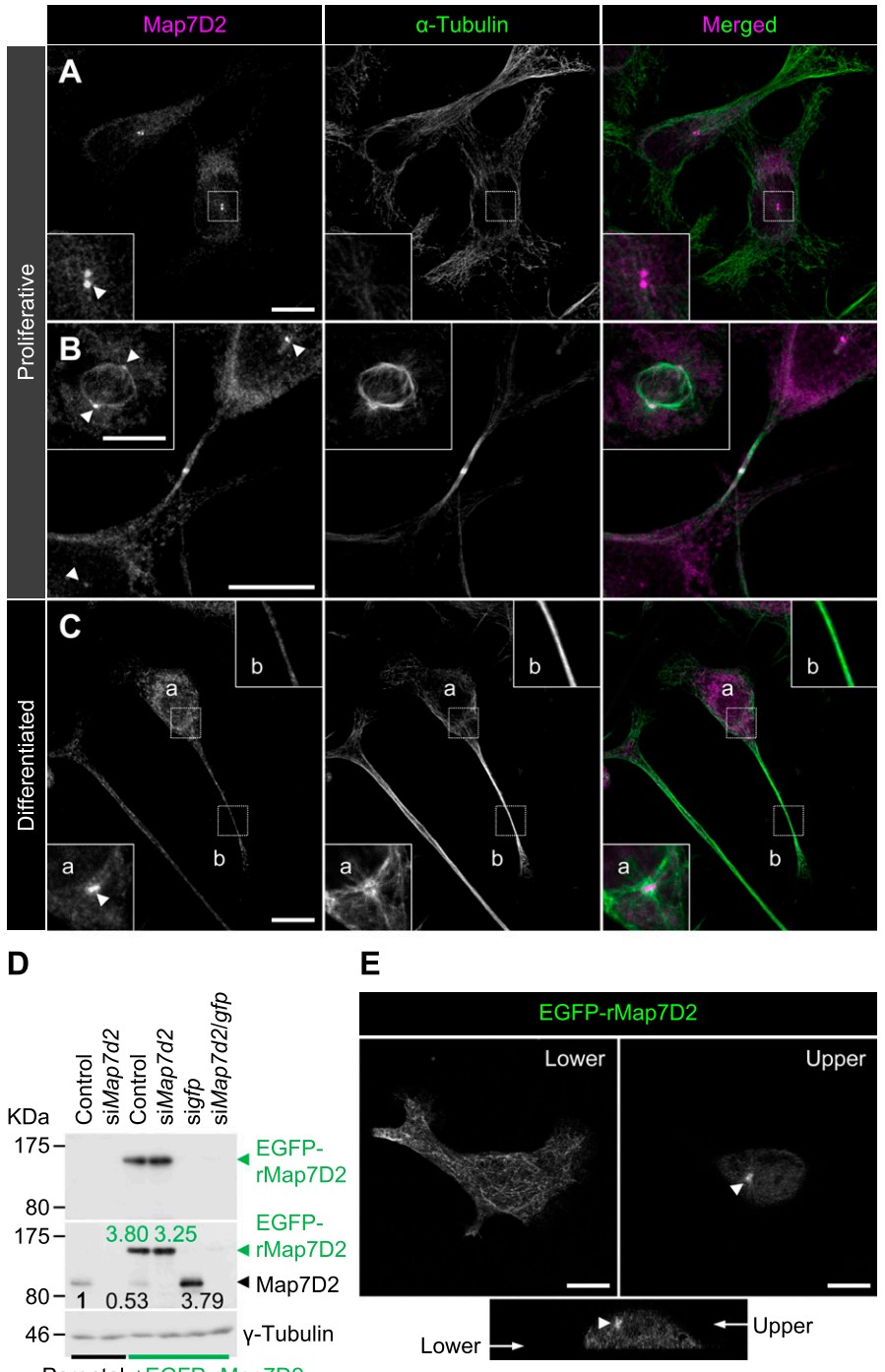

**Figure 3. Subcellular localization of Map7D2 in proliferative and differentiated N1-E115 cells.**
**(A, B, C)** Localization of Map7D2 in interphase (A), mitosis (B), and differentiation state (C) of N1-E115 cells. Cells were double-stained with anti-Map7D2 and anti–α-tubulin antibodies. In (A), the insets show enlarged images of regions indicated by a white box. In (B), the inset shows metaphase cells. In (C), images of differentiated cells were captured by z-sectioning because the focal planes of the centrosome and neurites are different. Each inset shows an enlarged image of the region indicated with a white box at each focal plane. Arrowheads indicate the centrosomal localization of Map7D2. **(D)** Generation of N1-E115 cells stably expressing EGFP-rMap7D2. To check the expression level of EGFP-rMap7D2, lysates derived from the indicated cells were probed with anti-GFP (top panel) and anti-Map7D2 (middle panel) antibodies. The blot was reprobed for γ-tubulin as a loading control (bottom panel). The amount of endogenous Map7D2 or EGFP-rMap7D2 was normalized to the amount of γ-tubulin, and the value relative to endogenous Map7D2 in the parental control was calculated. **(E)** Confirmation for subcellular localization of Map7D2 using N1-E115 cells stably expressing EGFP-rMap7D2. Images were captured by z-sectioning. Top panels show images taken with the lower or upper focal plane, and bottom panels show the image reconstructed in the z-axis direction. Arrow head shows centrosomal localization of Map7D2. Data information: scale bars, 10 μm in (A, B, C, E).
Source data are available for this figure.

localize not only partially on MTs but also strongly to the centrosome in N1-E115 cells (Figs 3 and S5B–D). The centrosome acts as a microtubule-organizing center that contributes to MT nucleation, stabilization, and anchoring (Sanchez & Feldman, 2017). Knockdown of *Map7d2* or *Map7d1* resulted in the increased MT shrinkage in response to a low concentration of nocodazole (Fig 4A), clearly indicating that Map7D2 and Map7D1 are required for MT stabilization. On the other hand, the reduced intensity of α-tubulin

around the centrosome by knockdown of either may be possibly because of reduced MT stabilization and reduced MT nucleation or anchoring (Fig 5C and D). Therefore, the mechanisms regulating localization patterns of Map7D2 and Map7D1 may differ, and Map7D2 and Map7D1 may have other functions at the centrosome such as MT nucleation and anchoring, depending on the cell type.

We also found that among the stable MT marker acetylated and detyrosinated tubulins, *Map7d1* knockdown reduced only

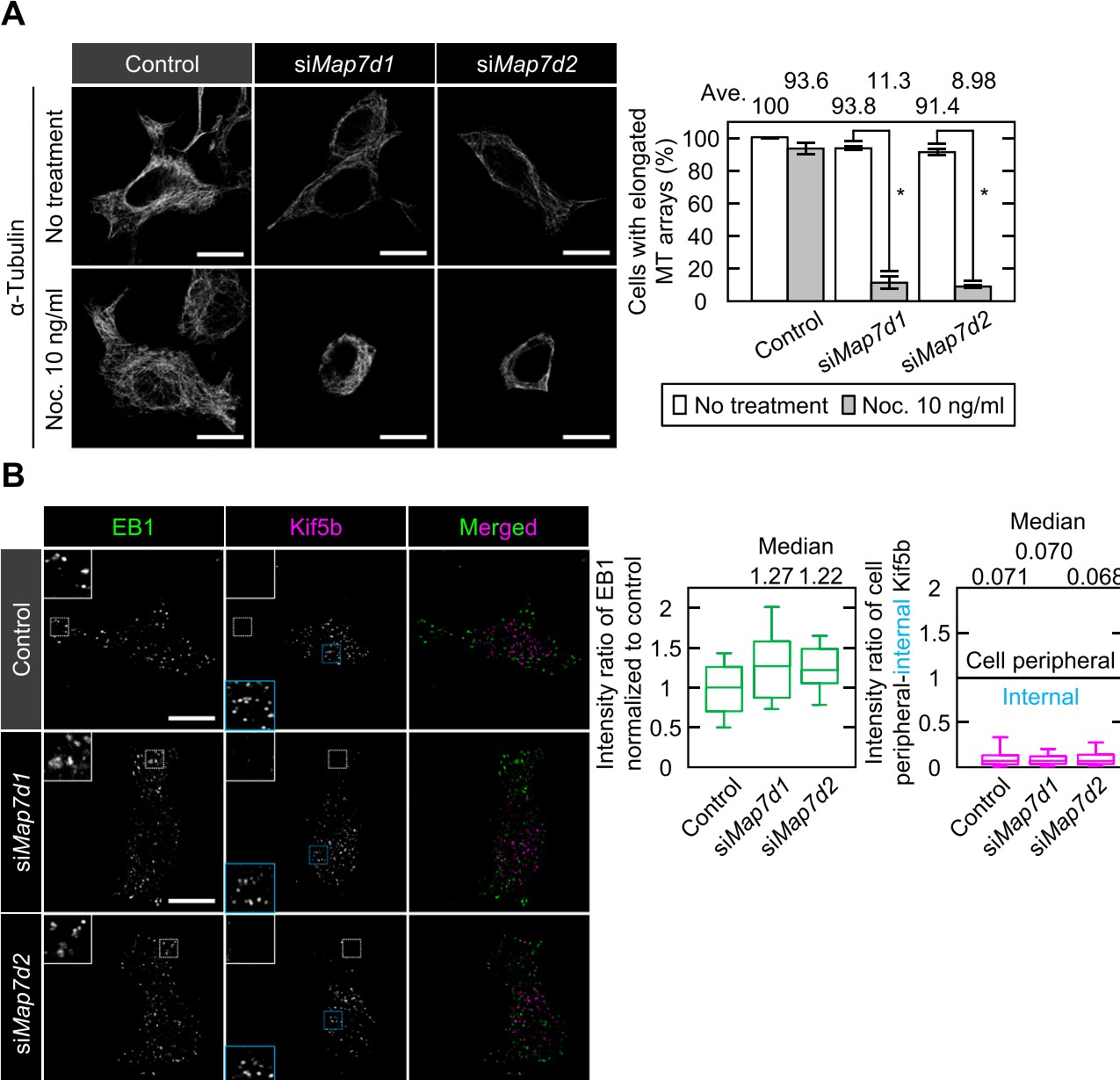

**Figure 4. Map7D2 is required for MT stabilization within the cell.**
**(A)** Confirmation of MT stability by low-dose nocodazole treatment in the indicated N1-E115 cells. The cells were treated with or without a low concentration of nocodazole (10 ng/ml) for 1 h and were stained with an anti–α-tubulin antibody. Cells with the elongated MT arrays or the MT shrinkage were counted, and the rate of cells with the elongated MT arrays was calculated. Data are from three independent experiments and represent the average ± SD. **(B)** Immunofluorescence staining for EB1 and Kif5b in N1-E115 cells treated with each siRNA (Top panels). The insets show enlarged images of regions indicated by a white box. Bottom left panel, the intensities of EB1 at the cell periphery in the indicated cells were measured via ROI analysis (each, n = 30 cells from three independent experiments). Data from *Map7d1* or *Map7d2* knockdown were shown by normalizing with the control value. Bottom right panel, the intensities of Kif5b at the cell periphery (white box) and the internal region (cyan box) were measured via ROI analysis, and the intensity ratios of cell peripheral-internal Kif5b were calculated (each, n = 30 cells from three independent experiments). Of note, a value of 1 means that Kif5b is distributed throughout the cell, and a value greater or less than 1 means that Kif5b is distributed at the cell periphery or in the internal region, respectively. Data information: In (A), *P < 0.0008 (the *t* test). Scale bars, 10 μm in (A, B).
Source data are available for this figure.

acetylated tubulin (Fig 5A, C, and D). These two modifications are mediated by different mechanisms and are sometimes not synchronous (Yoshiyama et al, 2003; Sudo & Baas, 2010; Song & Brady, 2015; Bance et al, 2019). For instance, Montagnac et al have shown

that defects in the α-tubulin acetyltransferase αTAT1-Clathrin–dependent endocytosis axis reduce only acetylated tubulin, resulting in a shift from directional to random cell migration (Montagnac et al, 2013). In addition, acetylated tubulin is down-

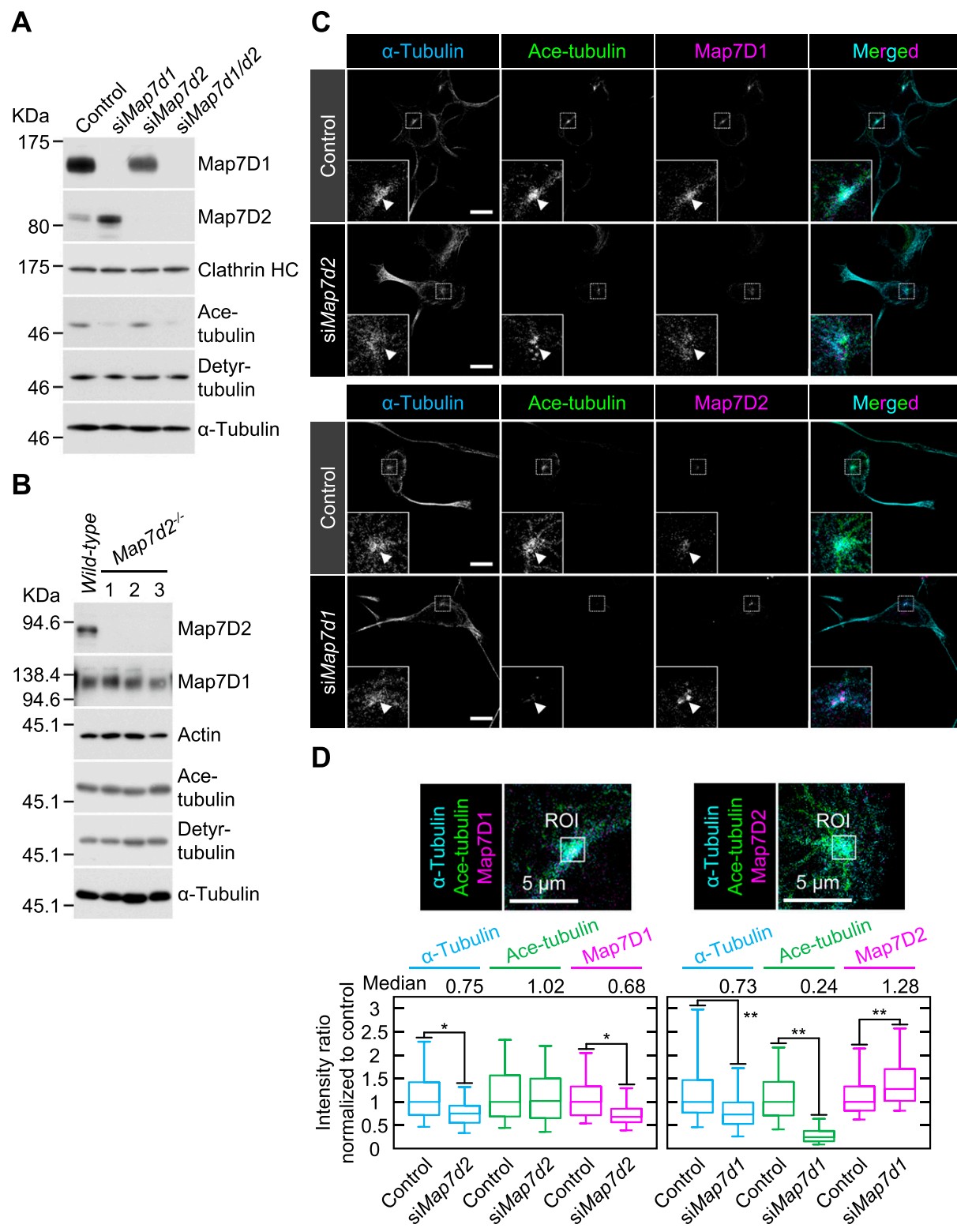

**Figure 5. Map7D2 and Map7D1 facilitate MT stabilization through distinct mechanisms.**
**(A)** Immunoblot analysis for acetylated (Ace-) and detyrosinated (Detyr-) tubulin in N1-E115 cells treated with each siRNA. Lysates derived from the indicated cells were separated by SDS–PAGE and subjected to immunoblotting with anti-Map7D1, anti-Map7D2, anti–Ace-tubulin, or anti–Detyr-tubulin antibodies. The blot was reprobed for Clathrin heavy chain (HC) or α-tubulin as a loading control. **(B)** Immunoblot analysis for Ace- and Detyr-tubulins in *wild-type* and *Map7d2*−/− N1-E115 cells. Three independent *Map7d2*−/− clones were used in this study. Lysates derived from the indicated cells were separated by SDS–PAGE and were immunoblotted with anti-Map7D1, anti-Map7D2, anti–Ace-tubulin, or anti–Detyr-tubulin antibodies. The blot was reprobed for α-actin or α-tubulin as a loading control. **(C)** Immunofluorescence

regulated by tubulin deacetylases such as HDAC6 and Sirt2 (Hubbert et al, 2002; North et al, 2003). Therefore, it is conceivable that Map7D1 may be involved in the pathway that controls the level of acetylated tubulin.

We also determined the tissue distribution of Map7D2, which has not been described to date (Fig 1). Among the MAP7 family members, the tissue distribution of Map7 has been analyzed in detail so far (Fabre-Jonca et al, 1998; Komada et al, 2000). At the mRNA level, *Map7* is expressed in a variety of epithelial tissues, dorsal root ganglia, trigeminal ganglia, and primitive seminiferous tubules during embryonic development. We also reported that both Map7 and Map7D1 are expressed in the epithelia of the mouse fallopian tube at the protein level (Kikuchi et al, 2018). Consistent with Map7 expression in primitive seminiferous tubules, *Map7* homozygous gene-trap mice exhibited defects in spermatogenesis (Komada et al, 2000). Map7D2 was expressed predominantly in the glomerular layer of the olfactory bulb and Sertoli cells of the testis (Fig 1C). The glomerular layer is known to be the region where axons accumulate and does not express Map2, a marker of neuronal cell bodies and dendrites (Fig 1C). As Map7D2 localizes to the proximal axon in cultured hippocampal neurons (Pan et al, 2019), Map7D2 may have similar localization and function in olfactory bulb neurons. The function of Map7D2 in Sertoli cells was not clarified in the present study. Therefore, whether Map7D2 is involved in mammalian neurogenesis and spermatogenesis represents a question for future research.

# Materials and Methods

### Molecular cloning, expression, and purification of rMap7D2

Based on the information of DDBJ/EMBL/GenBank accession number NM_001289778.1, oligonucleotide primers (5′-ATGTCGA-CATGGAGCGCAGCGGTGGGAACGGCG-3′ and 5′-ATGTCGACTCAACA-GAAGGTGTTCAGCGTAGTTTC-3′) were designed, and rat *Map7d2* (r*Map7d2*) cDNA was obtained by PCR using rat cDNA as a template. Expression vectors for r*Map7d2* were constructed in pCMV5-Myc (Nakanishi et al, 1997), pQE9 (QIAGEN), pGEX-5X-3 (Cytiva), pcDNA3.1/V5-His (Thermo Fisher Scientific), pCLXSN-GFP (Reiley et al, 2005), and pEGFP-N3 (Clontech). GST-fused proteins were expressed in *Escherichia coli* and purified using glutathione-Sepharose beads (Cytiva), respectively. GST-rMap7D2 (full length) was further purified by gel filtration using a HiLoad 16/60 Superdex 200 column (Cytiva).

### Antibodies

A rabbit polyclonal anti-Map7D2 antibody was raised against GST-rMap7D2 (aa 1–235). All the primary antibodies used are listed in Table S1. Secondary antibodies coupled to HRP were purchased from Sigma-Aldrich. Alexa Fluor–conjugated secondary antibodies used for immunofluorescence experiments were purchased from Thermo Fisher Scientific.

### MT binding assay

The MT co-sedimentation assay was performed as previously described (Yamamoto et al, 2002), with a slight modification. MTs were prepared by incubating tubulin in polymerization buffer (80 mM PIPES/NaOH [pH 6.8], 1 mM $MgCl_2$, 1 mM EGTA, and 1 mM GTP) containing 10% glycerol for 20 min at 37°C. After incubation, Taxol was added at a final concentration of 15 $\mu M$. Various amounts of rMap7D2 were incubated with 0.4 mg/ml of MTs in polymerization buffer containing 15 $\mu M$ taxol for 20 min at 37°C. After incubation with MTs, the mixture (200 $\mu l$) was placed over a 700-$\mu l$ cushion of 50% sucrose in polymerization buffer containing 15 $\mu M$ taxol. After the sample was centrifuged at 100,000$g$ for 30 min at 37°C, the supernatant was removed from the cushion, and the original volume was restored with SDS sample buffer. Comparable amounts of the supernatant and pellet fractions were subjected to SDS–PAGE, followed by Coomassie Brilliant Blue (CBB) protein staining. The amount of protein was estimated using a densitometer. ELISA for MT binding was performed in a 96-well microtiter plate as previously described (Pedrotti et al, 1994). Briefly, wells were coated by incubating with 0.2 mg/ml of MTs in polymerization buffer containing 15 $\mu M$ taxol for 2 h at 37°C and then blocked via incubation with 5% glycine. Increasing amounts of rMap7D2 were added to each well and incubated for 20 min at 37°C. The plate was washed and further incubated with an anti-Map7D2 antibody. After washing, the plates were incubated with a secondary antibody conjugated to HRP. SuperSignal ELISA Pico (Pierce) was used as a chemiluminescent peroxidase substrate.

### MT polymerization assays

MT assembly was assayed by measuring turbidity at 350 nm using a spectrophotometer, as previously described (Gaskin et al, 1974). Briefly, GST-rMap7D2 (0.14 mg/ml) was incubated with 2 mg/ml tubulin in polymerization buffer at 37°C. The sample was continuously monitored at 350 nm using a Hitachi U-2000 spectrophotometer. MT assembly was further assayed by fluorescence microscopy using rhodamine-labeled tubulin (Hyman et al, 1991). Briefly, GST-rMap7D2 (0.07 mg/ml) was incubated at 37°C for 20 min with 0.8 mg/ml tubulin (1:9 = rhodamine-labeled tubulin:unlabeled tubulin) in polymerization buffer. Incubation was stopped through the addition of 1% glutaraldehyde. The sample was spotted onto a glass slide and viewed under a fluorescence microscope.

---

staining for $\alpha$-tubulin, Ace-tubulin, and Map7D1 or Map7D2 in N1-E115 cells treated with each siRNA. For a comparison of signal intensities, images were captured under the same parameters. The insets show enlarged images of regions indicated by a white box. Of note, Ace-tubulin was present predominately around the centrosome in N1-E115 cells as indicated by arrowheads. **(C, D)** Quantification for immunofluorescence staining shown in (C). Left panels, the intensities of $\alpha$-tubulin, Ace-tubulin, and Map7D1 around the centrosome in the indicated cells were measured via ROI analysis (control, n = 197 cells; si*Map7d2*, n = 192 cells from three independent experiments). Right panels, the intensities of $\alpha$-tubulin, Ace-tubulin, and Map7D2 around the centrosome in the indicated cells were measured by ROI analysis (control, n = 193 cells; si*Map7d1*, n = 227 cells from three independent experiments). Data information: In (D), the bars of box-and-whisker plots show the 5 and 95 percentiles. *$P < 1 \times 10^{-13}$; **$P < 1 \times 10^{-8}$ (the $t$ test). Scale bars, 10 $\mu m$ in (C) and 5 $\mu m$ in (D).
Source data are available for this figure.

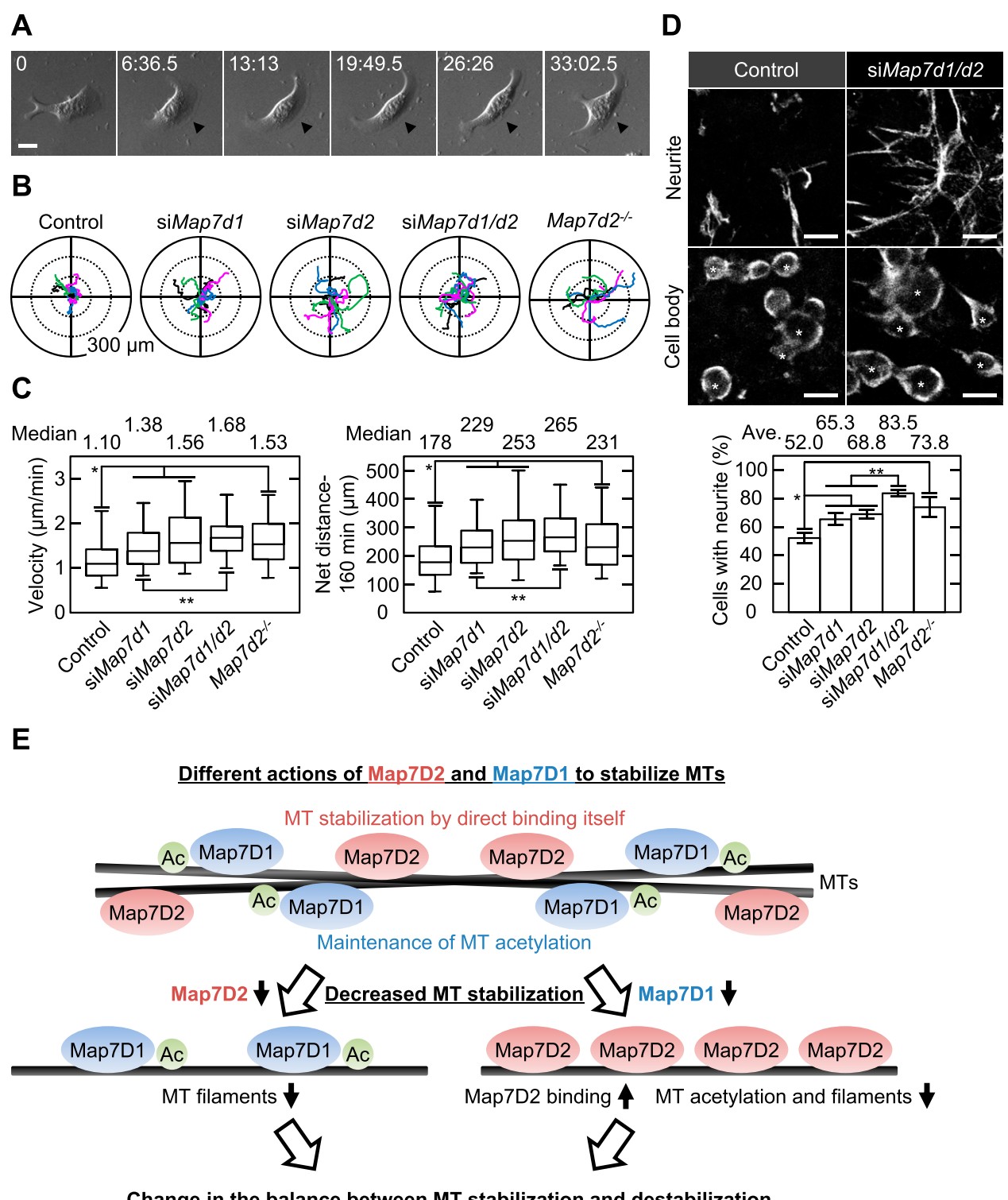

**Figure 6. Map7D2 suppresses random cell migration and neurite outgrowth.**
**(A)** Bright-field images of migrating N1-E115 cells. Arrowheads show lamellipodia formed in the direction of migration. **(B)** Tracking analysis of random cell migration in the indicated cells. Each color represents the trajectory of 12 randomly selected cells. **(C)** Velocity and net distance measured in the indicated cells (control: n = 114 cells; si*Map7d1*: n = 100 cells; si*Map7d2*: n = 71 cells; si*Map7d1/d2*: n = 107 cells; *Map7d2*$^{-/-}$: n = 60 cells from three independent experiments). **(D)** Neurite outgrowth assay in the indicated cells. Neurites and cell bodies were visualized by α-tubulin staining (upper). The neurite outgrowth from each cell was distinguished by acquiring images with Z-sectioning. Data are from three or four independent experiments and represent the average ± SD. **(E)** Proposed model for the distinct mechanisms of Map7D2 and

## Northern blotting

An RNA blot membrane (Clontech) was hybridized with the $^{32}$P-labeled full coding sequence of rMap7D2, according to the manufacturer's protocol.

## Cell culture and transfection

HeLa and N1-E115 cells were cultured at 37°C in DMEM supplemented with 10% fetal bovine serum and penicillin–streptomycin. The methods employed for plasmid or siRNA transfection were previously described (Kikuchi et al, 2010). Plasmid transfection of N1-E115 cells was performed using Lipofectamine LTX according to the manufacturer's instructions. Differentiation of N1-E115 neuroblastoma cells was induced by decreasing the serum level to 0.5% fetal bovine serum and adding 1% DMSO (hereafter, the above-described medium was referred to as differentiation medium). Stealth double-stranded RNA was purchased from Thermo Fisher Scientific. All siRNAs used in this study are listed in Table S2. Three individual siRNAs for mouse *Map7d2* or *Map7d1* were designed based on the respective sequences. Double-stranded RNA targeting luciferase was used as a control. The cells were cultured for 72 h and subjected to various experiments. In Figs 3D, 4A and B, 5A, C, D, and 6B–D, to exclude siRNA off-target effects, a mixture of three individual siRNAs for Map7D1 or Map7D2 was used. For the generation of N1-E115 cells stably expressing EGFP-rMap7D2, clones were selected by adding G418 at 24 h post-transfection. EGFP-rMap7D2 expression was confirmed by immunoblotting using antibodies against GFP and Map7D2. For Fig 4A, cells were treated with 10 ng/ml of nocodazole (Sigma-Aldrich) for 1 h.

## Generation of *Map7d2* knockout N1-E115 cell lines by CRISPR-Cas9

Two sgRNA sequences were designed using the CHOPCHOP CRISPR/Cas9 gRNA finder tool (http://chopchop.cbu.uib.no/). The short double-stranded DNA for each sgRNA (5′-CACCGTGAAGAGAGCA-CATGTGCC-3′ and 5′-AAACGGCACATGTGCTCTCTTCAC-3′, or 5′-CACC GCAGGATCACCAGGGCCTGG-3′ and 5′-AAACCCAGGCCCTGGTGATCCTGC-3′) was inserted into the *Bbs*I site of pX330 (Cong et al, 2013). To construct the *Map7d2* knockout vector, the 5′ and 3′ arms of each gene were amplified by PCR using N1-E115 genomic DNA and cloned into the pCR4 Blunt-TOPO vector (Thermo Fisher Scientific). The puromycin resistance marker was inserted between the 5′ and 3′ arms (Fig S6A). N1-E115 cells were transfected with 1 µg of each of the two pX330-sgRNA plasmids and the knockout vector using Lipofectamine LTX (Thermo Fisher Scientific). Knockout clones were selected by adding puromycin (Sigma-Aldrich) at 24 h post-transfection. Successful knockout was confirmed by immunoblotting using an anti-Map7D2 antibody and genomic PCR.

## Animals

Mice (C57BL6/N; Japan SLC) were used in this study. Animal care and experiments were conducted in accordance with the guidelines for the care and use of laboratory animals of the Center for Animal Resources and Development, Kumamoto University. All experiments were approved by the experimental animal ethics committee of Kumamoto University (A2019-127 and A2021-018). Mice were kept in a light- and temperature-controlled room with a 12-h light/dark cycle at 22°C ± 1°C.

## Quantitative real-time PCR

Each RNA sample was subjected to reverse transcription using murine leukemia virus reverse transcriptase (Thermo Fisher Scientific), and the generated cDNA was used as a template for qRT-PCR. Each reaction mixture was prepared using the KAPA SYBR Fast qPCR kit (Kapa Biosystems), and the PCR reaction was performed on ViiA7 (Thermo Fisher Scientific). The primers used for RT-qPCR are listed in Table S3.

## Immunoblotting and immunoprecipitation

For immunoblotting, cells were washed once with PBS and lysed with Laemmli's sample buffer. After boiling, the lysates were separated by SDS–PAGE, transferred to polyvinylidene difluoride membranes (Millipore), and immunoblotted with antibodies. For immunoprecipitation analysis, the HeLa cells were washed once with PBS at 24 h post-transfection and lysed with 1×NP40 buffer (20 mM Tris–HCl [pH 8.0], 10% glycerol, 137 mM NaCl, 1% NP40) supplemented with protease inhibitors and phosphatase inhibitors for 20 min on ice. The supernatant was collected after centrifugation and incubated with the appropriate antibodies. After incubation, 15 µl of protein A or G Sepharose beads was added, and the mixtures were rotated for 1 h at 4°C. The beads were washed once with 1×NP40 buffer, twice with LiCl buffer (0.1 M Tris–HCl [pH 7.5], 0.5 M LiCl), once with 10 mM Tris–HCl (pH 7.5), and were finally resuspended in Laemmli's sample buffer.

## Immunofluorescence and imaging analyses

For immunofluorescence staining, cells were grown on coverslips and fixed in 100% methanol at −20°C for 5 min. After blocking with 1% BSA in PBS for 1 h at room temperature, the samples were incubated with primary antibodies overnight at 4°C, followed by incubation with Alexa Fluor–conjugated secondary antibodies (Thermo Fisher Scientific) for 1 h. For immunofluorescence tissue staining, tissues were fixed in 4% paraformaldehyde in PBS at 4°C overnight and then immersed sequentially in 10, 20, and 30% sucrose in PBS at 4°C. After sucrose equilibration, tissues were immersed in OCT (Sakura Finetechnical) at room temperature for 5 min, followed by embedding in OCT and freezing in liquid nitrogen. Sections (10 µm) were stored at −80°C. The sections were washed once with PBS for 10 min and twice with 0.1% Triton X-100 in PBS for 10 min. After blocking with Blocking One (Nacalai) for 1 h at room temperature, the samples were incubated with primary antibodies

Map7D1 for MT stabilization. See the Discussion section for further detail. Data information: In (C), the bars of box-and-whisker plots show the 5 and 95 percentiles. *$P < 1 \times 10^{-4}$; **$P < 0.002$ (the t test). In (D), *$P < 0.002$; **$P < 0.0002$ (the t test). Scale bars, 20 µm in (A, D).
Source data are available for this figure.

overnight at 4°C, followed by incubation with Alexa Fluor–conjugated secondary antibodies (Thermo Fisher Scientific) for 1 h. Nuclei were stained with DAPI for 30 min at room temperature. The samples were viewed under a fluorescence microscope (BX51; Olympus) or a confocal microscope (FV1000; Olympus or TCS SP8; Leica). Images were processed and analyzed using Fiji software (National Institutes of Health).

### Random cell migration assay and neurite outgrowth assay

For the random cell migration assay, cells were seeded onto a laminin-coated (10 μg/ml) glass-bottom dish and recorded under an inverted microscope system equipped with an incubator (LCV110; Olympus). For the neurite outgrowth assay (Fig S7B), the underside of 3-μm pore transwell membranes (Corning) was coated with 500 μl of 10 μg/ml laminin in PBS into a well of a 24-well plate. After coating, the membranes were removed from the laminin and placed into the well of a 24-well dish containing 500 μl differentiation media. A total of 100 μl of cell suspension (containing 1–2 × $10^5$ cells) was added to the insert chamber on top of the membrane. The cells were allowed to extend neurites through the membrane pores to the lower chamber (underside of the membrane) for 6 h at 37°C. The cells were then fixed and stained with an anti–α-tubulin antibody. Images were processed and analyzed using Fiji software (National Institutes of Health).

### Statistics

The experiments were performed at least three times (biological replicates), and the results are expressed as the average ± SD or the median, first, and third quartiles and 5–95% confidence intervals for the box-and-whisker plot. Differences between data values, except for Fig 2D, were tested for statistical significance using the $t$ test. Statistical significance was set at $P < 0.05$. In Fig 2D, differences between data values were tested for statistical significance using the F-test.

### Other procedures

Tubulin was prepared from fresh porcine brains by three cycles of polymerization and depolymerization, followed by DEAE-Sephadex column chromatography (Shelanski et al, 1973; Williams & Lee, 1982).

## Data Availability

Source data for each figure are available in PDF or Excel format. In this study, we do not deposit any data in public databases.

## Supplementary Information

## Acknowledgments

We thank past and present members of our laboratory for helpful discussions. This work was partly carried out at the Institute of Molecular Embryology and Genetics, the Gene Technology Centre, the Center for Animal resources and Development, and Research Facilities of the School of Medicine, Kumamoto University. This work was supported by JSPS KAKENHI Grant Numbers 22700881, 24700980, 15K07054, and 19K06664, and grants from The Japan Spina Bifida and Hydrocephalus Research Foundation, Astellas Foundation for Research on Metabolic Disorders, the Mochida Memorial Foundation, the Takeda Science Foundation, and the Uehara Memorial Foundation (to K Kikuchi).

### Author Contributions

K Kikuchi: conceptualization, resources, data curation, formal analysis, supervision, funding acquisition, validation, and investigation, visualization, methodology, project administration, and writing—original draft, review, and editing.
Y Sakamoto: resources, formal analysis, and writing—review and editing.
A Uezu: resources and formal analysis.
H Yamamoto: formal analysis.
K-i Ishiguro: resources and methodology.
K Shimamura: methodology.
T Saito: methodology.
S-i Hisanaga: methodology.
H Nakanishi: resources and formal analysis.

### Conflict of Interest Statement

The authors declare that the research was conducted in the absence of any commercial or financial relationships that could be construed as a potential conflict of interest.

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
