## [Reviewer comments · Life Science Alliance]

Life Science Alliance

Map7D2 and Map7D1 facilitate microtubule stabilization through distinct mechanisms in neuronal cells

Koji Kikuchi, Yasuhisa Sakamoto, Akiyoshi Uezu, Hideyuki Yamamoto, Kei-ichiro Ishiguro, Kenji Shimamura, Taro Saito, Shin-ichi Hisanaga, and Hiroyuki Nakanishi

DOI: <https://doi.org/10.26508/lsa.202201390>

Corresponding author(s): *Dr. Koji Kikuchi (Kumamoto University)*

Review Timeline:	Submission Date:	2022-01-27
	Editorial Decision:	2022-01-31
	Revision Received:	2022-03-01
	Editorial Decision:	2022-03-29
	Revision Received:	2022-04-01
	Accepted:	2022-04-01

Scientific Editor: Novella Guidi

Transaction Report:

Please note that the manuscript was reviewed at Review Commons and these reports were taken into account in the decision-making process at Life Science Alliance.

January 31, 2022

Re: Life Science Alliance manuscript #LSA-2022-01390

Dr. Koji Kikuchi
Graduate School of Medical Sciences, Kumamoto University
Department of Molecular Pharmacology
1-1-1 Honjo
Kumamoto, Kumamoto 860-8556
Japan

Dear Dr. Kikuchi,

Thank you for submitting your manuscript entitled "Map7D2 and Map7D1 facilitate microtubule stabilization through distinct mechanisms to control cell motility and neurite outgrowth." to Life Science Alliance. The manuscript was submitted and reviewed via Review Commons. The authors then chose to transfer their somewhat revised manuscript, along with the reviewers' comments and a proposed revised plan to Life Science Alliance (LSA). The reviewer comments and revision plan was assessed at LSA, and LSA editors deemed that the manuscript could be further considered at LSA provided the authors revise the manuscript, in accordance to what they have laid out in the pbp rebuttal / revision plan.

We, thus, encourage you to submit a revised manuscript to us that includes all the experiments you have laid out in their Revision plan. Given that new data will be added to the revised manuscript, the revision will have to be looked at by a set of referees, most likely the same ones as Review Commons.

Thank you for this interesting contribution to Life Science Alliance. We are looking forward to receiving your revised manuscript.

Sincerely,

B. MANUSCRIPT ORGANIZATION AND FORMATTING:

Reviewer #1**Evidence, reproducibility and clarity (Required):**

In this manuscript, Kikuchi et al describe the characterization of MAP7D2 and MAP7D1, two MAP7 family members in mouse with specific expression patterns. Focusing mostly on MAP7D2, they assess its expression pattern across the body and find that it is mostly expressed in certain neuronal subsets. They then characterize the MT-related properties of MAP7D2 based on previous knowledge of other MAP7 family members. They show that MAP7D2 binds MTs (via the N-terminus), determine the binding affinity, and show that it can stimulate MT polymerization (or stabilization) both in vitro and in vivo. Using a specific antibody, they localize MAP7D2 to centrosomes, midbody and neurites in N1-E115 cells. Functionally, they show that loss of MAP7D1/2 mildly affects microtubule stability as judged by acetyl-tubulin staining, and properties of these cells that rely on cytoskeletal elements such as cell migration and neurite growth. Interestingly, there might be a feedback loop regulating MAP7D1/2 expression, as knockdown of MAP7D1 upregulates MAP7D2.

Overall, the experiments and conclusions are very solid and convincing, such that I would not ask for further experiments. This is in part because the experiments are largely based on previous characterizations of other MAP7 family members, which are **largely confirmed**. **The presentation of the data is also very clear.**

Significance (Required):

*I see the value of the study in the fact that **it provides solid and specific research tools** for MAP7D1/2 which could be **very useful** for the microtubule/neuronal cytoskeleton community.*

Response: We thank the reviewer very much for appreciating the content of our manuscript.

****Referees cross-commenting****

Reviewers 2 and 3 criticize that the evidence for an effect of MAP7D1/2 on MT dynamics is weak. I would agree in that ac-tub stainings and in vitro experiments are rather indirect. The experiments suggested by reviewer 2 should clarify this (esp. nocodazole should be easy). I also agree that an experiment addressing the potential involvement of kinesin-1 would help, the involvement of which seems to have been omitted by the authors. A kinesin-binding deficient mutant would add another MAP7D1/2 tool and increase the value for the community.

Response: As for the reviewer's suggestions listed above, please refer to our responses to the comments of Reviewer #2.

Reviewer #2**Evidence, reproducibility and clarity (Required):**

In this study, the authors investigate 2 members from the MAP7 family Map7D2 and Map7D1. They first address the tissue distribution of Map7D2, by northern blotting using a variety of rat tissues. To complement their analysis, they also raised an antibody to look at the protein distribution. From their

studies, they concluded that Map7D2 is abundantly expressed in the brain and testis. The authors went on to perform a series of functional assays. First, they biochemically demonstrated that rat Map7D2 directly binds to MTs by MT co-sedimentation assay. The MT binding domain was mapped to the N-terminal half. They performed MT turbidity assay to demonstrate enhanced MT polymerisation in the presence of Map7D2, suggesting that this Map stabilises MTs. The authors went on to characterise in detail the subcellular localisation of Map7D2 which was predominantly present in the centrosome and partially localised to MTs including within neurites from N1-E115 cells. Kikuchi et al. further revealed the overlap in expression between Map7D2 and another family member, Map7D1. The authors continued these studies by a series of functional studies in N1-E115 cells where they performed single or combined knock-downs of Map7D2 and Map7D1 and studied the levels of acetylated and detyrosinated tubulins and the effect of the knock-downs on migration and neurite extension. The main conclusion from this work was that Map7D2 and Map7D1 facilitate MT stabilization through distinct mechanisms which are important in controlling cell motility and neurite outgrowth. Map7D2 is proposed to stabilise MTs by direct binding whereas Map7D1 does it indirectly by affecting acetylation.

Major comments:

The main conclusion from this work that Map7D2 and Map7D1 facilitate MT stabilization and that this is necessary for correct migration and neurite extension has not been convincingly demonstrated. In my opinion, a more detailed study of MT properties to demonstrate a role in MT stabilisation would greatly benefit the work, eg. experiments using MT destabilising agents such as nocodazole. In addition, a series of experiments aiming to study MT dynamics would help to understand the function of these MT regulators. The authors proposed an elevation in microtubule dynamics to explain the increase in migration and neurite extension but no experimental proof was provided.

Response: According to the reviewer's suggestion, we assessed the role of MT stabilization in greater detail by analyzing the effects of *Map7d2* or *Map7d1* knock-down on the resistance to the MT-destabilizing agent, nocodazole. In control N1-E115 cells, a small degree of MT shrinkage was observed, and elongated MT arrays were retained even after treatment with a low concentration of nocodazole (10 ng/mL) for 1 h (new Fig 4A). In contrast, not only *Map7d2* but also *Map7d1* knock-down dramatically increased MT shrinkage when subjected to nocodazole treatment at the same concentration (new Fig. 4A), indicating that both Map7D2 and Map7D1 are required for MT stabilization within the cell. We have described these new findings shown in new Fig. 4A in the Results section. (page 8, lines 10–17)

To study MT dynamics, methods such as analyzing the velocity and direction of an EB1-GFP comet are commonly used. We have previously analyzed the roles of Map7 and Map7D1 in MT dynamics using HeLa cells stably expressing EB1-GFP (Kikuchi et al., *EMBO Rep.*, 2018). However, no such tools have been developed for analyzing MT dynamics in N1-E115 cells, which were used in this study. In addition, it is difficult to analyze MT dynamics by transient expression of EB1-GFP because of the low plasmid transfection efficiency. Therefore, we tried to assess the effect on MT dynamics by measuring the EB1 comet length by immunofluorescence, referring to Fig. 7D in *EMBO J.* 32:1293–1306, 2013. However, it was hard to measure the EB1 comet length in N1-E115 cells, because the size of the EB1 comet observed in N1-E115 cells was much smaller than that in other cells such as HeLa cells. Instead, we measured the amount of EB1-decorated MTs at the cell periphery based on the intensity of EB1. As shown in new Fig. 4B, the knock-down of either *Map7d2* or *Map7d1* did not affect the intensity of EB1 at the cell periphery, compared to control N1-E115 cells, suggesting that both Map7D2 and Map7D1 are

dispensable for the elongation of EB1-decorated MTs. In addition, since Map7D2 and Map7D1 can form a complex with Kif5b (new Fig. S4) (Kikuchi et al., *EMBO Rep.*, 2018; *Cell Rep.*, 26: 1988-1999, 2019.), we also examined whether *Map7d2* or *Map7d1* knock-down affects the distribution of Kif5b foci. As shown in new Fig. 4B and new Fig. S6A, even after the knock-down of *Map7d2* or *Map7d1*, the distribution of Kif5b foci was similar to that in control N1-E115 cells. Kif5b foci were predominantly located at the internal regions of the cell (new Fig. 4B), and some were partly observed in the protrusions (new Fig. S6A). We have described these new findings shown in new Fig. 4B and new Fig. S6A in the Results section. (page 8, lines 17–26)

Moreover, considering the possibility that the Map7D2 dynamics are altered when MT stability is changed, e.g., before and after differentiation induction, we analyzed the Map7D2 dynamics at the centrosome by fluorescence recovery after photobleaching (FRAP) using N1-E115 cells stably expressing EGFP-rMap7D2. As shown in new Fig. 4C and new Fig. S6B, we found that the dynamics were altered between the proliferative and differentiated states. Compared to the proliferative state, the recovery rate of EGFP-rMap7D2 was reduced (new Fig. 4C), and the immobile fraction of Map7D2 was increased in the differentiated state (new Fig. S6B). These data suggest that the increase in immobile Map7D2 may enhance MT stabilization. We have described these data shown in new Fig. 4C and new Fig. S6B in the Results section. (page 8, line 29 to page, 9 line 4)

It has been previously demonstrated that loss of MAP7D2 leads to a decrease in axonal cargo entry to axons resulting in defects in axon development and neuronal migration. The C-terminus is necessary for this function as it mediates interaction with Kinesin-1 (Pan et al., 2019). Such mechanisms could also explain the defects in migration and neurite growth that the authors observed. This possibility has not been considered but instead, the subtle changes in total α -tubulin led to suggest MT stabilisation as a key function without proof of causation. Could the authors provide some further experimental evidence to demonstrate that stability is the main contributor to the phenotypes observed? Eg. by rescuing migration and neurite phenotypes with a variant of MAP7D2 which cannot bind kinesin1.

Response: The reviewer states “*Such mechanisms could also explain the defects in migration and neurite growth that the authors observed;*” however, our results showed that loss of Map7D2 elevated the rates of both cell motility and neurite outgrowth (original Fig. 5). In contrast, it has been reported in several papers that when Kinesin-1 function is impaired, both cell motility and neurite outgrowth are reduced (*Curr. Biol.*, 23: 1018–1023, 2013; *Mol. Cell. Biol.*, 39: e00109–19, 2019; etc.). In addition, we assessed the effects of *Map7d2* or *Map7d1* knock-down on the distribution of kif5b foci. Even after the knock-down of either, the distribution of kif5b foci was still similar to that in control N1-E115 cells, as shown in new Fig. 4B. Although it is possible that in *Map7d2* or *Map7d1* knock-down N1-E115 cells, the effects of reduced MT stabilization are offset by those of Kinesin-1 dysfunction, resulting in a mild increase in the rate of cell motility and neurite outgrowth, the phenotypes we observed are likely independent of the functions associated with Kif5b in N1-E115 cells. It is indeed possible that the experiment suggested by the reviewer may reveal relationships between Map7D2 and Kinesin-1 in terms of cell motility and neurite outgrowth, however, it is difficult to conduct such an experiment because transient expression of Map7D2 induces MT bundling, as shown in original Fig. 2F. Based on the above, we have added a discussion on the relationship between Map7D2 and Kinesin-1. (page 10, line 29 to page 11, line 7)

A key conclusion proposed by the authors is that Map7D2 and Map7D1 facilitate MT stabilization through distinct mechanisms. Such different roles in MT stabilisation are important in controlling cell motility and neurite outgrowth. In my opinion, their data does not fully support this statement and the findings using MT readouts do not match the defects in migration and neurite growth. Loss of Map7D2 leads to a very subtle phenotype on α -tubulin, while Map7D1 decreases both α -tubulin and acetylated tubulin, but Map7D1 seems to have a milder or similar effect on migration and neurite growth than Map7D2. Furthermore, it would be expected that the combined loss of function would lead to a stronger phenotype in cell migration when compared to the single loss of functions due to their distinct roles on MT stability, however, this seems not to be the case.

Response: The fact that no stronger phenotype was observed may be because, besides Map7D2 and Map7D1, other molecules are involved in MT stabilization. Another possible explanation is that the increases in both cell motility and neurite outgrowth caused by decreased MT stabilization are offset by Kinesin-1 dysfunction. We have added a discussion on the above two possibilities. (page 11, lines 4–9)

Minor comments:

1) *In the first result section, the author refers to Fig. S3 to suggest the expression of MAP7D2 in the cerebral cortex, however, there are no transcripts in the cerebral cortex according to the figure. Similarly, the immunofluorescence analysis done by the authors shows marginal expression of MAP7D2 in the cerebral cortex.*

Response: According to the reviewer's comment, we have changed the order of the data shown in new Fig. 1C, top panels. The data from the olfactory bulb, cerebellum, and hippocampus, in which Map7D2 expression was detected in the database, were arranged in the top three rows, and the data from the cerebral cortex, in which Map7D2 expression was not detected in the database, were moved to the bottom row as a negative control. In addition, we have revised the relevant part of the Results section. (page 5, lines 10–17)

2) *The authors use γ -Tubulin as a housekeeping gene in Fig. 3D, since Map7D2 is enriched in centrosomes this may not be the most appropriate choice.*

Response: γ -Tubulin is abundant in both the cytosol and the nuclear compartments of cells (Sig. Transduct. Target Ther. 3: 24, 2018). As it has been used for similar purposes in several other studies (Cancer Res., 61: 7713–7718, 2001; J. Biol. Chem., 291: 23112–23125, 2016; etc.), we considered it acceptable for use as a loading control for immunoblotting.

3) *According to the authors, knockdown of Map7D2 leads to a decrease in the intensity of α -tubulin and Map7D1 (Fig. 4C and D). This data doesn't agree with the previous statement made by the authors where they show that Map7D2 knockdown or knockout did not affect Map7D1 expression by Western Blot Analysis (Fig. S2C and S5B)*

Response: The immunoblotting results indicate that the total amount of Map7D1 in the cells is not affected by loss of Map7D2. In contrast, the immunofluorescence results indicate that the amount (distribution) of Map7D1 localized around the centrosome is decreased by loss of Map7D2, presumably

due to a reduction in the number of MT structures that can serve as scaffolds for Map7D1. We have added this interpretation in the Results section. (page 9, lines 16–19)

4) Line 6 page 7 "Endogenous Map7D2 expression is suppressed in N1-E115 cells stably expressing EGFP-rMap7D2 and was restored by specific knock-down of EGFP-rMap7D2 using *gfp* siRNA (Fig. 3D)". No quantifications and stats are shown. Also, endogenous Map7D2 after knock-down of EGFP-rMap7D2 is not comparable to the control.

Response: According to the reviewer's suggestion, we have quantified the amount of endogenous Map7D2 or EGFP-rMap7D2, normalized it to the amount of γ -tubulin, and calculated relative values to endogenous Map7D2 in the parental control. The amount of endogenous Map7D2 was decreased to 53% in N1-E115 cells stably expressing EGFP-rMap7D2, suggesting that EGFP-rMap7D2 expression suppressed endogenous Map7D2 expression. In this cell line, the total amount of Map7D2 (EGFP-rMap7D2 + endogenous Map7D2) was increased, however, when EGFP-rMap7D2 was depleted using *sigfp* in this cell line, endogenous Map7D2 was expressed to the same level as EGFP-rMap7D2 before knock-down. Together with the finding that *Map7d1* knock-down increased the amount of Map7D2, these findings indicate that the amount of Map7D2 in the cells is regulated in response to the amount of Map7D1 and exogenous Map7D2. We have added this interpretation in the Results section. (page 7, lines 24–31)

In addition, we have revised the legend of new Fig. 3D to clarify the quantification method. (page 27, lines 21–23)

5) Line 8 page 7 "These results suggest that the expression of Map7D2 was influenced by changes in that of Map7D1" This statement seems in the wrong place, after the Map7D2 and EGFP-rMap7D2 experiment. Instead for clarity, it would be better placed after line 5 where the authors explain the effect of Map7D1 knock-down on the levels of Map7D2.

Response: According to the reviewer's suggestion, we have rephrased the relevant sentence as follows: "Interestingly, *Map7d1* knock-down upregulated Map7D2 expression, as confirmed with three different siRNAs (Fig. S2C), suggesting that Map7D2 expression is affected by changes in Map7D1 expression, not by off-target effects of a particular siRNA." (page 7, lines 23, 24)

6) Line 8 page 8 "Although the physiological role of the C-terminal region of Map7D2 is currently unknown..." This statement seems not adequate as there are several studies reporting the role of the C-terminal region of Map7D2 in Kinesin1- mediated transport. The authors mention such studies in the discussion.

Response: According to the reviewer's suggestion, we have added a discussion of the relationship between Map7D2 and kinesin-1. (page 10, lines 10–16, and page 10, line 29 to page 11, line 7)

7) Line 6 page 9 " Further, the knock-down of either resulted in a comparable reduction of MT intensity (Fig. 4C and D) ..." This is not visible and/or justified by the images provided and would benefit from some sort of quantification at other regions such as neurites.

Response: Considering the cell motility, quantification of α -tubulin/Ace-tubulin/Map7D1/Map7D2 intensities in neurites is not appropriate. Instead, we have added arrowheads indicating α -tubulin/Ace-tubulin/Map7D1/Map7D2 in new Fig. 5C for better understanding.

8) In Fig. 2B, a band corresponding to his6-rMAP7D2 of molecular weight >97 kDa co-sedimented with the microtubules. However, the cloned rMAP7D2 had a molecular weight of 84.82 kDa and the addition of 6XHis-Tag would add another 2-3 kDa, therefore, the final protein band observed should be less than 90 kDa. It would be beneficial if the authors could specify the molecular weight of the purified protein after the addition of the V5-his tag and/or if there was addition of amino acids due to cloning strategy.

Response: In Fig. 2B, we used full-length GST-tagged rMap7D2, like in Fig. 2E and D; therefore, we have corrected His₆-rMap7D2 as GST-rMap7D2. We apologize for the mistake.

9) In Fig. 2C, there is misalignment of the western blot with the panel or text underneath.

Response: We thank the reviewer for pointing this out; we have corrected the misalignment of the CBB staining in new Fig. 2C.

10) In Fig. 3C the inset from the first panel seems to correspond to a different focal plane than the main image.

Response: We have revised the relevant part of the figure legend as follows: “In C, images of differentiated cells were captured by z-sectioning, because the focal planes of the centrosome and neurites are different. Each inset shows an enlarged image of the region indicated with a white box at each focal plane. Arrowheads indicate the centrosomal localization of Map7D2.” (page 27, lines 15–18)

11) In Fig. 4A, the cell type is not specified and is referred as “indicated cells”, also the material and methods section seems to omit the specific cells used.

Response: We have added “in N1-E115 cells treated with each siRNA” in the legend of new Fig. 5A.

12) Fig. S6 is not mentioned in the results.

Response: We apologize for having referred to original Fig. S6 only in the Discussion section in the original manuscript. We have described the findings shown in the original Fig. S6 in the Results section and have renumbered the figure as new Fig. S4. (page 6, lines 18–22)

Significance (Required):

MTs play essential roles in practically every cellular process. Their precise regulation is therefore crucial for cellular function and viability. MAPs are specialised proteins that interact with MTs and regulate their behaviour in different manners. Understanding their precise function in different cellular contexts is of utmost importance for many biological and biomedical fields.

MAPs are well known for their ability to promote MT polymerization, bundling and stabilisation in vitro (Bodakuntla et al., 2019). Several members of the Map7 family have been shown to regulate microtubule stability. For instance, MAP7 can prevent nocodazole-induced MT depolymerization and maintain stable microtubules at branch points in DRG neurons (Tymanskyj & Ma, 2019). Ensconsin, the *Drosophila* Map, is required for MT growth in mitotic neuroblasts by regulating the mean rate of MT polymerization (Gallaud et al., 2014). However, this family of Maps seems to have diverse functions encompassing a variety of mechanisms, as exemplified by a series of studies demonstrating the involvement of MAP7 family proteins in the recruitment and activation of kinesin1 (Hooikaas et al., 2019; Pan et al., 2019) and in microtubule remodelling and Wnt5a signalling (Kikuchi et al., 2018). Further understanding of this family of Maps and how its members differ in their function is important and will help to advance the field.

Response: We appreciate the reviewer's comments. We believe that our revision plan will greatly improve the quality of our manuscript.

Reviewer #3

Evidence, reproducibility and clarity (Required):

Summary:

Microtubule Associated Proteins (MAPs) are important regulators of microtubule dynamics, microtubule organization and vesicular transport by modulating motor protein recruitment and processivity. In the current manuscript the authors have characterized 2 members of the MAP7 protein family, MAP7D1 and MAP7D2. The authors characterized MAP7D2 expression pattern in the brain and its microtubule binding properties in vitro and in cells. In cells both proteins localize to the centrosome and to microtubules and upon depletion centrosome localized microtubules seem reduced, and cell migration and neurite outgrowth are increased. Surprisingly, they find that microtubule acetylation (a common marker for stable microtubules) is reduced upon MAP7D1 depletion but not MAP7D2 depletion. Based on this finding the authors conclude that these proteins have a distinct mechanism in stabilizing MTs to affect cell migration and neurite outgrowth; MAP7D2 stabilizes by binding to MTs, whereas MAP7D1 stabilizes MTs by acetylation.

Main comments:

- Both MAP7 proteins show strong localization to the centrosome and to a lesser degree to MTs. Knockdown of either protein leads to reduced MTs around the centrosome, which lead the authors to conclude the MAP7s are stabilizing the MTs. However, the effect could just as well be an indirect effect due to a function of these MAPs at the centrosome. To address this authors could e.g. quantify microtubule properties in postmitotic cells. In addition, antibody specificity should be tested using knockdown of knockout cells, as this centrosome localization was not observed in HeLa cells (Hooikaas, 2019; Kikuchi, 2018). Maybe this localization is specific to rat MAP7s or to the cell line used.

Response: We presume that this comment partly overlaps with the comments by Reviewer #2. We assessed the role of MT stabilization in greater detail by analyzing the sensitivity to an MT-destabilizing agent, nocodazole, and the effect on MT elongation by measuring the intensity of EB1 by immunofluorescence. Please refer to our responses to the comments of Reviewer #2.

Regarding the reviewer's concern about antibody specificity, we had carefully confirmed the antibody specificity, as shown in original Fig. S2. Subsequently, Map7D2 localization was confirmed in N1-E115 cells stably expressing EGFP-rMap7D2, as shown in original Fig. 3D, E. In addition, we are currently conducting analyses using *Map7d1-egfp* knock-in mice, which confirmed that Map7D1 localizes around the centrosome in cortical neurons, as shown below (**we would like to disclose these unpublished data to the reviewers only**). Therefore, it is thought that the localization pattern of Map7D2 and Map7D1 differs depending on the cell type. **We have added this interpretation to the Discussion section. (page 7, lines 17–21)**

[Figure Removed by Life Science Alliance editorial staff]

- *Centrosome nucleated microtubules are typically highly dynamic and little modified. Therefore is the Ac-tub staining at the centrosome really MTs? I cannot identify MTs in the fluorescent images in 4C. Maybe authors could consider ac-tub/alpha-tub ratio in non centrosomal region (e.g. neurites). Moreover, as both Acetylation and deetyrosination are associated with long-lived/stable MTs, it is surprising that only acetylated tubulin goes down on WB. Does this suggest that long-lived MTs are still present to normal level? If so, can one still argue that the loss of acetylation is the cause of the lower MT levels? This should at least be discussed.*

Response: As for the reviewer's statement "*Centrosome nucleated microtubules are typically highly dynamic and little modified. Therefore is the Ac-tub staining at the centrosome really MTs?*", it has been previously reported that tubulin acetylation is observed around the centrosome in some cell lines (*J. Neurosci.*, 30: 7215–7226, 2010; *PLoS One*, 13: e0190717, 2018; etc.). N1-E115 is one of the cell lines in which tubulin acetylation is observed around the centrosome.

It is not surprising that "*only acetylated tubulin goes down on WB,*" as it has been previously reported that acetylated and deetyrosinated tubulins are sometimes not synchronous (*J. Neurosci.*, 23: 10662–10671, 2003; *J. Neurosci.*, 30: 7215–7226, 2010; *J. Cell Sci.*, 132: jcs225805, 2019., etc.). For instance, Montagnac et al. (*Nature*, 502: 567–570, 2013) showed that defects in the α -tubulin acetyltransferase α TAT1-Clathrin-dependent endocytosis axis reduce only tubulin acetylation, resulting in a shift from directional to random cell migration. Although the details of the molecular function of Map7D1 are beyond the main purpose of this study, **we have added a discussion on reduced tubulin acetylation by *Map7d1* knock-down based on the above. (page 11, lines 12–21)**

- MAP7D1 and MAP7D2 depletion leads to subtle defect in cell migration and neurite outgrowth, which the author suggest is caused by reduced MT stability. However, MAP7 proteins have well characterized functions in kinesin-1 transport, and thus the phenotypes may well be caused by defects in kinesin-1 transport. Ideally the authors would do rescue experiments with FL or just the MT binding N-termini to separate these functions. Moreover this is needed to substantiate the claim of the authors that MAP7D1 effect on MT stability is not mediated by direct binding.

Response: As this comment largely overlaps with the comments raised by Reviewer #2, please refer to our responses to the comments of Reviewer #2.

- The authors do not refer well to published work. Several papers have published very similar work (especially to Fig1+2) and it would help the reader much if this would be discussed/compared along the results section and not briefly mention these in the results section. In addition, authors overstate the novelty of their results e.g. page 3: these proteins are not "functionally uncharacterized" nor are their expression patten and biochemical properties analyzed for the first time in this manuscript; page 8 "Although the physiological role of the C-terminal region of Map7D2 is currently unknow, ..." There is a clear function for the C-terminus for the recruitment/activation of kinesin-1.

Response: According to the reviewer's suggestion, we have added a comparison with data on Map7 family members presented in previous papers in the Results section. This has already been mentioned in the discussion section. (page 5, lines 5–8, 17-19, and 21)

Furthermore, we have rephrased the relevant part describing the physiological role of the C-terminal region of Map7D2. (page 6, lines 18–22)

Minor comments

- P6 Map7D3 also binds with its N-terminus to MTs, like other MAP7s (Yadav et al)

Response: According to the reviewer's comment, we have revised this as "Map7D3 binds through a conserved region on not only the N-terminal side, but also the C-terminal side (Sun, 2011; Yadav et al., 2014)." (page 6, lines 12–14)

- P7 "As Map7D2 has the potential to functionally compensate for Map7D1 loss" where is this based on?

Response: For clarity, we have rephrased this as "As Ma7D2 expression was upregulated upon suppression of Map7D1 expression, Map7D2 has the potential to functionally compensate for Map7D1 loss." (page 8, line 3)

- Fig2F quality of black-white images is low potentially due to conversion issues

Response: We thank the reviewer for pointing out these conversion issues, and we have made the necessary corrections.

Significance (Required):

At this stage the conceptual advance is limited. Part of the findings are not novel. The finding that MAP7s depletion have a different effect on MTs acetylation may be interesting to cytoskeleton researchers, although the potential mechanism has not been addressed experimentally or textually. However, their conclusion that this leads to reduced MTs and then to cell migration and neurite formation defects is not sufficiently supported by experimental evidence.

Response: We appreciate the reviewer's comments. We believe that our revision plan will greatly improve the quality of our manuscript.

****Referees cross-commenting****

I completely agree with reviewer #2: At this stage the paper's conclusions are not sufficiently supported by the data. Important will be to further characterize the effect on the MTs (do they really have a different effect) and to look at the possible involvement of the motor recruitment. Maybe that a 3 to 6 months revision time would have been more accurate.

Response: Please refer to our responses to the comments of Reviewer #2.

March 29, 2022

RE: Life Science Alliance Manuscript #LSA-2022-01390R

Dr. Koji Kikuchi
Kumamoto University
Department of Molecular Pharmacology
1-1-1 Honjo
Kumamoto, Kumamoto 860-8556
Japan

Dear Dr. Kikuchi,

Thank you for submitting your revised manuscript entitled "Map7D2 and Map7D1 facilitate microtubule stabilization through distinct mechanisms in neuronal cells". We would be happy to publish your paper in Life Science Alliance pending final revisions necessary to meet our formatting guidelines.

- please address the remaining Reviewer 1 points
- please add the Twitter handle of your host institute/organization as well as your own or/and one of the authors in our system
- the RNA seq datasets link in figure S3 does not work. Please fix it.

A. FINAL FILES:

B. MANUSCRIPT ORGANIZATION AND FORMATTING:

**Submission of a paper that does not conform to Life Science Alliance guidelines will delay the acceptance of your

manuscript.**

The license to publish form must be signed before your manuscript can be sent to production. A link to the electronic license to publish form will be sent to the corresponding author only. Please take a moment to check your funder requirements.

Sincerely,

Reviewer #1 (Comments to the Authors (Required)):

In general, the authors did an ok job at addressing issues from the reviewers issues. Importantly the conclusions are better supported by evidence, such as e.g. the title. I do feel it is a pity the authors did not look at MT dynamics. If transfection is really this difficult, the authors could have used e.g. Sir-Tubulin. Additionally, I still think that some of my issues were not well addressed and should (at least) textually be discussed.

Point 1: MAP7D1 and D2 show a very strong enrichment to the centrosome. This is probably not by MT binding (looks very different to MT labeling), but I expect to be a more direct binding to some PCM protein. What could be the role there? Could D1/D2 be involved centrosome integrity or in MT nucleation? And if so, could changes in "cytoplasmic" MTs densities be a consequence of a centrosome defect.

The FRAP experiment: I do not follow / don't agree the experiment and conclusion. This also relates to the point above. If MAP7D2 is centrosomal, this is potentially not MT bound protein. Therefore I do not see how this data can be used to conclude something about the MT cytoskeleton. Moreover I agree that there is a small difference in recovery speed but the final plateau is the same. Therefore there is no change in immobile fraction. I would argue for removing the FRAP data.

Reviewer #1 (Comments to the Authors (Required)):

In general, the authors did an ok job at addressing issues from the reviewers issues. Importantly the conclusions are better supported by evidence, such as e.g. the title. I do feel it is a pity the authors did not look at MT dynamics. If transfection is really this difficult, the authors could have used e.g. Sir-Tubulin. Additionally, I still think that some of my issues were not well addressed and should (at least) textually be discussed.

Point 1: MAP7D1 and D2 show a very strong enrichment to the centrosome. This is probably not by MT binding (looks very different to MT labeling), but I expect to be a more direct binding to some PCM protein. What could be the role there? Could D1/D2 be involved centrosome integrity or in MT nucleation? And if so, could changes in "cytoplasmic" MTs densities be a consequence of a centrosome defect.

The FRAP experiment: I do not follow / don't agree the experiment and conclusion. This also relates to the point above. If MAP7D2 is centrosomal, this is potentially not MT bound protein. Therefore I do not see how this data can be used to conclude something about the MT cytoskeleton. Moreover I agree that there is a small difference in recovery speed but the final plateau is the same. Therefore there is no change in immobile fraction. I would argue for removing the FRAP data.

Response: We thank the Reviewer for your suggestions. Regarding the Reviewer's concern about MT dynamics, we have toned down the statement in Fig. 6E as follows: "Change in the balance between MT stabilization and destabilization, resulting in increased random cell migration and neurite outgrowth".

In response to the Reviewer's comment concerning the localization of Map7D2 and Map7D1 to the centrosome, we have added a discussion on the possibility of Map7D2 and Map7D1 functioning at the centrosome. (page 11, lines 9–20)

Furthermore, according to the reviewer's suggestion, we have removed the FRAP data from our revised manuscript.

April 1, 2022

RE: Life Science Alliance Manuscript #LSA-2022-01390RR

Dr. Koji Kikuchi
Kumamoto University
Department of Molecular Pharmacology
1-1-1 Honjo
Kumamoto, Kumamoto 860-8556
Japan

Dear Dr. Kikuchi,

Thank you for submitting your Research Article entitled "Map7D2 and Map7D1 facilitate microtubule stabilization through distinct mechanisms in neuronal cells". It is a pleasure to let you know that your manuscript is now accepted for publication in Life Science Alliance. Congratulations on this interesting work.

DISTRIBUTION OF MATERIALS:

Again, congratulations on a very nice paper. I hope you found the review process to be constructive and are pleased with how the manuscript was handled editorially. We look forward to future exciting submissions from your lab.

Sincerely,
